# Structure of *Escherichia coli* respiratory complex I reconstituted into lipid nanodiscs reveals an uncoupled conformation

Piotr Kolata[1,2], Rouslan G Efremov[1,2]*

[1]Center for Structural Biology, Vlaams Instituut voor Biotechnologie, Brussels, Belgium; [2]Structural Biology Brussels, Department of Bioengineering Sciences, Vrije Universiteit Brussel, Brussels, Belgium

**Abstract** Respiratory complex I is a multi-subunit membrane protein complex that reversibly couples NADH oxidation and ubiquinone reduction with proton translocation against transmembrane potential. Complex I from *Escherichia coli* is among the best functionally characterized complexes, but its structure remains unknown, hindering further studies to understand the enzyme coupling mechanism. Here, we describe the single particle cryo-electron microscopy (cryo-EM) structure of the entire catalytically active *E. coli* complex I reconstituted into lipid nanodiscs. The structure of this mesophilic bacterial complex I displays highly dynamic connection between the peripheral and membrane domains. The peripheral domain assembly is stabilized by unique terminal extensions and an insertion loop. The membrane domain structure reveals novel dynamic features. Unusual conformation of the conserved interface between the peripheral and membrane domains suggests an uncoupled conformation of the complex. Considering constraints imposed by the structural data, we suggest a new simple hypothetical coupling mechanism for the molecular machine.

*For correspondence:
rouslan.efremov@vub.vib.be

**Competing interests:** The authors declare that no competing interests exist.

## Introduction

Complex I, NADH:ubiquinone oxidoreductase, is a multi-subunit enzyme found in many bacteria and most eukaryotes. It facilitates transfer of two electrons from NADH to ubiquinone, or its analogues, coupled reversibly with translocation of four protons across the membrane against trans-membrane potential (*Galkin et al., 2006*; *Sazanov, 2015*). Structures of the complete complex I from several eukaryotes (*Fiedorczuk et al., 2016*; *Hunte et al., 2010*; *Kampjut and Sazanov, 2020*; *Zhu et al., 2016*), one thermophilic bacterium (*Baradaran et al., 2013*), and the partial structure of the membrane domain of *Escherichia coli* complex I (*Efremov and Sazanov, 2011*), have been determined.

The composition of complex I differs significantly between species. Mitochondrial complex I has a molecular weight 1 MDa and comprises more than 35 subunits (*Wirth et al., 2016*), whereas bacterial analogues are much smaller with molecular weight approximately 500 kDa. Complex I from all characterized species contains homologues of 14 core subunits; seven subunits each assemble into peripheral and membrane arms, joined at their tips and form the complex with a characteristic L-shape.

The peripheral arm, exposed to the cytoplasm in bacteria or the mitochondrial matrix in eukaryotes, contains binding sites for NADH, ubiquinone, and flavin mononucleotide (FMN) as well as eight or nine iron-sulfur clusters, seven of which connect the NADH and ubiquinone-binding sites (*Sazanov, 2015*) enabling rapid electron transfer (*Verkhovskaya et al., 2008*).

The membrane-embedded arm includes a chain of three antiporter-like subunits, NuoL, NuoM, and NuoN (*E. coli* nomenclature is used for the subunits hereafter) (*Efremov and Sazanov, 2011*), which are also found in the Mrp family of multisubunit $H^+$/Na antiporters (*Steiner and Sazanov, 2020*). Each antiporter-like subunit contains two structural repeats comprising five transmembrane helices (TMH, TMH4-8, and TMH9-13). TMH7 and TMH12 are interrupted by an extended loop in the middle of the membrane and the helix TM8 at the interface between symmetric motifs is interrupted by the π-bulge (*Baradaran et al., 2013*; *Efremov and Sazanov, 2011*). Membrane-embedded NuoH mediates interaction with the peripheral arm and also contains five-helix structural repeat found in antiporter-like subunits (*Baradaran et al., 2013*). Together with subunits NuoB and NuoD it forms an extended ubiquinone-binding cavity (Q-cavity) that stretches from the hydrophobic region of the membrane bilayer to the binding site of the ubiquinone head group (Q-site) found in the proximity of the terminal iron-sulfur cluster N2 (*Baradaran et al., 2013*).

The membrane arm features a continuous chain of conserved and functionally important ionizable residues positioned in the middle of the membrane. These are suggested to be involved in proton translocation and its coupling to electron transfer (*Baradaran et al., 2013*; *Efremov and Sazanov, 2011*). Attempts to visualize conformational changes in the membrane domain (*Kampjut and Sazanov, 2020*; *Parey et al., 2018*) have revealed rotation of the cytoplasmic half of TMH3 of NuoJ in mammalian complex I (*Agip et al., 2018*) and were associated with active-deactive transition. Recently, proton translocation mechanisms without conformational changes in antiporter-like subunits were suggested (*Kampjut and Sazanov, 2020*; *Steiner and Sazanov, 2020*). However, all proposed coupling mechanisms remain largely speculative and require further validation by functional, biochemical, and structural methods.

*E. coli* complex I is among the best functionally characterized complex I homologues. It has been studied using many biophysical and biochemical techniques (*Verkhovskaya and Bloch, 2013*). Combined with the possibility of fast and extensive mutagenesis (*Pohl et al., 2007*; *Verkhovskaya and Bloch, 2013*), it represents a highly attractive system to study the coupling mechanism. However, owing to its fragile and dynamic nature (*Verkhovskaya and Bloch, 2013*), high-resolution structures of this complex remain limited to a partial structure of the membrane domain (*Efremov and Sazanov, 2011*).

Here, we present a single-particle cryo-EM structure of the entire *E. coli* complex I reconstituted into lipid nanodiscs, with the peripheral arm structure solved at 2.1 Å resolution and that of the membrane domain at 3.7 Å.

## Results

### Overall structure

Twin-strep tag was added to genomically encoded subunit NuoF using a CRISPR-Cas9 based system (*Jiang et al., 2015*; *Figure 1—figure supplement 1*). This enabled single-step purification of solubilized complex (*Figure 1—figure supplement 2A*), which was further reconstituted into lipid nanodiscs comprising *E. coli* polar lipids and membrane scaffold protein MSP2N2 (*Figure 1—figure supplement 2A,B*). Mass photometry indicated that reconstituted complex I was homogeneous and monodisperse (*Figure 1—figure supplement 2C*).

Reconstituted complex I was active in catalyzing NADH:ubiquinone-1 (Q1) and NADH:decylubiquinone (DQ) redox reactions (*Figure 1—figure supplement 2D,E*). Both NADH:Q1 and NADH:DQ activities were sensitive to Q-site-specific inhibitor piericidin-A but were lower than those of the detergent-purified protein supplemented with lipids (*Sazanov et al., 2003*) likely reflecting reduced solubility of hydrophobic electron acceptors in the absence or at low concentration of detergent. While without compartmentalization the proton translocation activity cannot be assessed, the lipid environment provided by the nanodisc is expected to mimic closely lipid vesicles in which reconstituted purified *E. coli* complex I was shown to pump protons (*Steimle et al., 2011*).

We determined the single-particle cryo-EM structure of the reconstituted complex (*Figure 1*, *Figure 1—figure supplements 3* and *4*, *Table 1*, *Video 1*). Multiple conformations of the complex that differed by relative positions of the peripheral and membrane arms were revealed by 3D classification (*Figure 1—figure supplements 4* and *5*). Three conformations of the entire complex were reconstructed to average resolutions between 3.3 and 3.7 Å (*Figure 1—figure supplement 4*)

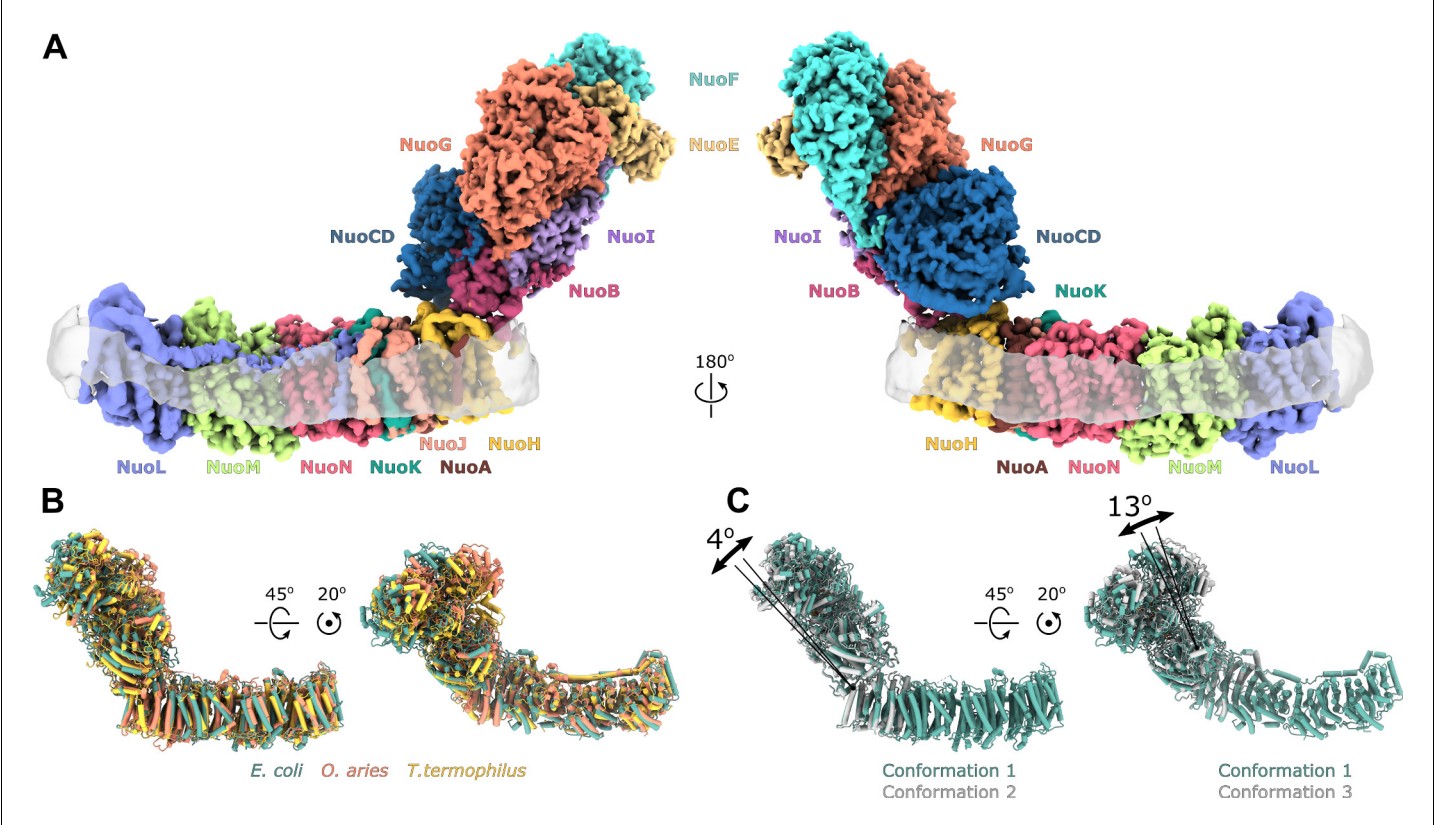

**Figure 1.** Architecture of *Escherichia coli* respiratory complex I. (**A**) Segmented density map of the complete complex I shown together with the nanodisc density (transparent gray). (**B**) Comparison of the structures of the *E. coli* (green), *Thermus thermophilus* (PDB ID: 4HEA, yellow), and the core subunits of ovine (PDB ID: 6ZKD, orange) complex I. (**C**) Conformational differences between three conformations resolved at high resolution. The structures are aligned on the membrane arm. The rotation axes and angles are indicated.

The online version of this article includes the following figure supplement(s) for figure 1:

**Figure supplement 1.** Schematic representation of Crispr-Cas9-enabled incorporation of the twin-strep tag into the N-terminus of the genomically encoded NuoF subunit.

**Figure supplement 2.** Purification and biochemical characterization of *E.coli* complex I reconstituted into lipid nanodiscs.

**Figure supplement 3.** Image processing diagram of the complex I in lipid nanodiscs.

**Figure supplement 4.** Properties of the nanodisc-reconstituted cryo-EM sample and the final reconstructions.

**Figure supplement 5.** Dynamic connection between peripheral and membrane arms.

**Figure supplement 6.** Representative cryo-EM density maps.

resolving the interface between the arms; however, due to high-residual mobility of the arms, the antiporter-like subunits were resolved at below 8 Å (*Figure 1—figure supplement 4*).

Focused refinement of each arm separately and subtraction of nanodisc density (*Figure 1—figure supplement 3*) improved the resolution of peripheral and membrane arms to 2.7 Å and 3.9 Å, respectively (*Figure 1—figure supplements 3* and *4*, *Table 1*). After density modification (*Terwilliger et al., 2020*), the resolution of membrane arm improved to 3.7 Å.

Micrograph analysis, in contrast to mass photometry, revealed that large fraction of the particles corresponds to the peripheral arm only (*Figure 1—figure supplement 3*) that may have dissociated during cryo-EM sample preparation. These yielded 3D reconstruction to 2.8 Å resolution (*Figure 1—figure supplement 3*), similar to the map of the peripheral arm of intact complex I. Joining two subsets and applying density modification improved resolution of the peripheral arm to 2.1 Å (*Table 1*, *Figure 1—figure supplements 4* and *6*). Using the resulting maps, atomic models of the peripheral and membrane arms have been built. The entire *E. coli* complex I was modeled by fitting models of the arms and extending additionally resolved loops and termini. Due to limited resolution, the antiporter-like subunits were refined as rigid bodies. The final models include 4618 residues that account for 94.7% of the total polypeptide constituting the complex (*Table 2*).

**Table 1.** Statistics of cryo-EM data collection, data processing, and model refinement.

**Data collection**

| | Nanodiscs | LMNG |
|---|---|---|
| Microscope | JEOL CRYOARM300 | |
| Acceleration voltage [kV] | 300 | |
| Energy filter | In-column Omega energy filter | |
| Energy filter slit width [eV] | 20 | |
| Magnification | 60 000 x | |
| Detector | Gatan K3 | |
| Physical pixel size [Å] | 0.771 | 0.766 |
| Exposure time [s] | 3 | 3 |
| Number of frames | 61 | 60 |
| Total electron dose [$e^-/Å^2$] | 65 | 60 |
| Defocus range [μm] | 0.9–2.2 | 1.0–2.0 |
| Number of micrographs collected | 9122 | 13,084 |
| Total number of particles extracted | 1,256,734 | 1,469,948 |

**Data processing**

| | Entire complex in nanodiscs | | | Membrane domain | Peripheral domain | Entire complex in LMNG |
|---|---|---|---|---|---|---|
| | Conformation 1 | Conformation 2 | Conformation 3 | | | |
| PDB ID: | 7NYR | 7NYU | 7NYV | 7NYH | 7NZ1 | - |
| EMDB ID: | EMD-12653 | EMD-12654 | EMD-12655 | EMD-12652 | EMD-12661 | EMD-13291 |
| Imposed symmetry | C1 | C1 | C1 | C1 | C1 | C1 |
| Final number of particles | 23,445 | 21,620 | 21,234 | 37,441 | 286,384 | 7,962 |
| Final resolution, RELION, FSC=0.143 | 3.9 | 4.6 | 4.5 | 3.9 | 2.4 | 6.7 |
| Final resolution, RELION, FSC=0.5 | 6.0 | 7.9 | 7.4 | 4.4 | 2.8 | 8.7 |
| Sharpening B-factor, RELION [$Å^2$] | −67 | −126 | −116 | −70 | −52 | −195 |
| Final resolution, PHENIX resolve_cryo_em, FSC=0.143 | 3.3 | 3.8 | 3.7 | 3.7 | 2.1 | - |
| Final resolution, PHENIX resolve_cryo_em, FSC=0.5 | 3.8 | 4.6 | 4.4 | 4.2 | 2.5 | - |
| Local resolution range (Å) | 2.9–11 | 3.2–15 | 3.1–13 | 3.0–6.2 | 1.8–3.8 | 4.8–12 |
| **Model refinement** | | | | | | |
| Initial model | 4HEA, 3RKO | 4HEA, 3RKO | 4HEA, 3RKO | 4HEA, 3RKO | 4HEA | - |
| Refinement package | PHENIX 1.19.2, Real-space refinement | | | | | - |
| Model resolution at FSC=0.5 (Å) | 3.5 | 4.3 | 4.1 | 3.8 | 2.2 | - |
| Cross-correlation | | | | | | |
| Mask | 0.69 | 0.58 | 0.62 | 0.70 | 0.80 | - |
| Volume | 0.67 | 0.57 | 0.60 | 0.68 | 0.75 | - |
| Model composition | | | | | | |
| Non-hydrogen atoms | 35229 | 35229 | 35229 | 16167 | 19773 | - |
| Protein residues | 4618 | 4618 | 4618 | 2195 | 2361 | - |
| Waters | 0 | 0 | 0 | 0 | 1170 | - |
| Ligands | 11 | 11 | 11 | 0 | 11 | - |
| B-factors mean ($Å^2$) | | | | | | |
| Protein | 50 | 93 | 93 | 75 | 31 | - |
| Ligand | 32 | 67 | 74 | - | 33 | - |

*Table 1 continued on next page*

*Table 1 continued*

**Data collection**

| | Nanodiscs | | | | | LMNG |
|---|---|---|---|---|---|---|
| Waters | - | - | - | - | 21 | - |
| R.M.S. deviations | | | | | | |
| Bond lengths (Å) | 0.005 | 0.000 | 0.005 | 0.006 | 0.006 | - |
| Bond angles (°) | 0.875 | 0.860 | 0.850 | 0.943 | 0.937 | - |
| **Validation** | | | | | | |
| MolProbity score | 1.38 | 1.45 | 1.52 | 1.35 | 0.91 | - |
| Clashscore | 3.81 | 4.73 | 5.73 | 2.33 | 1.36 | - |
| Poor rotamers (%) | 0.82 | 0.82 | 0.85 | 1.10 | 0.98 | - |
| C-beta outliers (%) | 0 | 0 | 0 | 0 | 0 | - |
| CaBLAM outliers (%) | 2.02 | 1.96 | 2.07 | 2.05 | 1.60 | - |
| Ramachandran plot (%) | | | | | | |
| Favored | 96.63 | 96.72 | 96.68 | 95.54 | 97.81 | - |
| Allowed | 3.37 | 3.28 | 3.32 | 4.46 | 2.19 | - |
| Outliers | 0.00 | 0.00 | 0.00 | 0.00 | 0.00 | - |

The fold and arrangement of the *E. coli* complex I subunits is mainly similar to the structures of other complex I homologs. Somewhat high values of RMSD obtained for structural alignments of the entire complexes or its individual arms to *Thermus thermophilus* [RMSD 7.3 Å (4199 Cα) for entire complex, 4.3 Å (2070 Cα) for membrane arm, and 8.1 Å (2129 Cα) for peripheral arm] and ovine enzyme [8.5 Å (4007 Cα) for entire complex, 5.0 Å (1969 Cα) for membrane arm, and 8.4 Å (2038 Cα) for the peripheral arm] (*Figure 1B*) reflect long-range twisting and bending of arms observed between complex I from different species (*Agip et al., 2018*; *Baradaran et al., 2013*; *Vinothkumar et al., 2014*).

Comparison of *E. coli* complex I conformations reconstructed to better than 4 Å resolution revealed two modes of relative arm rotation (*Figure 1C*): (1) rotation around an axis that passes through the NuoH-NuoB interface and is tilted around 45 degrees out of the plane formed by the arms with an amplitude of at least 13 degrees, and (2) rotation around an axis parallel to the membrane and roughly perpendicular to the long axis of the membrane arm with an amplitude of approximately 4 degrees. Although relative arm movements were observed in mammalian (*Kampjut and Sazanov, 2020*; *Zhu et al., 2016*) and *T. thermophilus* complex I (*Gutiérrez-Fernández et al., 2020*), their amplitudes were smaller and movement directionality was less diverse. Despite significant relative arm movements, the structure of each arm was rigid and did not reveal different conformations apart from the specific local dynamics discussed below.

## Structure of the peripheral arm
### Architecture of the peripheral arm reveals a novel evolutionary strategy to stabilize the subcomplex
At an average resolution of 2.1 Å with the local resolution in the core reaching 2.0 Å (*Figure 1— figure supplement 4*) conformations of most side chains in the peripheral arm, positions of ions, and multiple water molecules were resolved

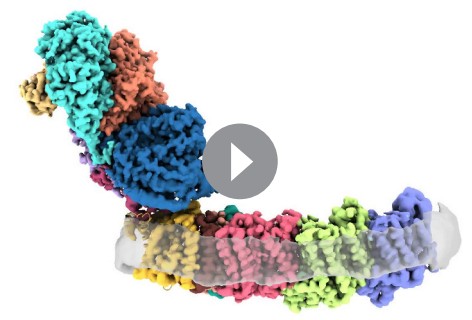

**Video 1.** Composite density map of *E. coli* complex I is shown along with density of the lipid nanodisc. The homology model of TMH1 is shown in ribbon representation.

https://elifesciences.org/articles/68710#video1

**Table 2.** Residues built in the models.

| Subunit | Total number of residues | Built residues | | | Co-factors | Fragments build in entire complex only |
|---|---|---|---|---|---|---|
| | | Entire complex | Membrane arm | Peripheral arm | | |
| NuoF | 445 | 1–441 | | 1–441 | FMN, N3 | |
| NuoE | 166 | 11–166 | | 11–166 | N1a | |
| NouG | 908 | 1–907 | | 1–907 | N1b, N4, N7 | |
| NuoI | 180 | 23–180 | | 39–180 | N6a, N6b | 23–38 |
| NuoB | 220 | 43–76, 86–179, 190–220 | | 53–71, 90–179, 190–220 | N2 | 43–53, 71–76, 86–90: |
| NuoCD | 596: C 1–172 D 212–596 | 9–596 | | 9–205, 210–218, 224–233, 238–596 | - | 206–209, 219–223, 234–237 |
| NuoH | 325 | 52–321 | 52–214, 223–321 | | - | 215–222 |
| NuoA | 147 | 15–38, 60–127 | 15–38, 66–127 | | - | 61–66 |
| NuoJ | 184 | 1–164 | 1–164 | | - | |
| NuoK | 100 | 1–100 | 1–100 | | - | |
| NuoN | 485 | 1–191, 199–437, 447–483 | 1–191, 199–437, 447–483 | | - | |
| NuoM | 509 | 1–504 | 1–504 | | - | |
| NuoL | 613 | 1–612 | 1–612 | | - | |

unambiguously (*Figure 2*, *Figure 1—figure supplement 6*).

Unlike other structurally characterized homologues, *E. coli* subunits NuoC and NuoD are joined in a single polypeptide. The 35 amino acid-long linker includes an α-helix (residues 180–194) that interacts with subunit NuoB (*Figure 2A*). The relative positions of all redox centers with FMN and nine iron-sulfur clusters, including off path cluster N7 (*Sazanov and Hinchliffe, 2006*), are particularly well conserved (*Figure 2C*).

A distinctive feature of the *E. coli* peripheral arm is the presence of ordered C-terminal extensions in subunits NuoB, NuoI, and NuoF with a length of 22–45 residues and a large 94 residue insertion loop in subunit NuoG, referred to as the G-loop (*Figure 2A*, *Table 3*). These extensions are unique among structurally characterized complex I homologues and have a well-defined structure. While the G-loop has a compact fold, the conformation of the C-terminal tails is extended. They line the surface of the peripheral arm subunits with high shape complementarity (*Figure 2A*, *Figure 2—figure supplement 1*) and apart from a few helical turns, have no secondary structure elements (*Table 3*). They create additional inter-subunit contacts with some surface areas exceeding 1000 Å$^2$ (*Table 3*). Similarly, the G-loop fills a crevice between NuoCD, NuoI, and NuoG subunits (*Figure 2A*) and together with the extensions increase the interaction surface between the electron acceptor module (NuoEFG) and connecting module (NuoICDB) by a factor of three (from 1400 to 4600 Å$^2$), thus stabilizing the peripheral arm assembly. These structural features are conserved within the Enterobacteriaceae family, are very common in the phylum Gammaproteobacteria and display high conservation of interfacial residues, particularly for the G-loop (*Figure 2—figure supplement 1*). They demonstrate a new evolutionary strategy for complex stabilization that was not observed in other complex I homologs.

A strong density near the NuoG surface coordinated by [G]Asp617, [G]Gln632, [G]Glu647, [G]Asp731 and four water molecules (*Figure 2B*) was assigned to a Ca$^{2+}$ ion. The coordination number, geometry, and ion-ligand distances of ca 2.5 Å (*Zheng et al., 2008*) as well as the 2 mM concentration of Ca$^{2+}$ in the buffer support this assignment. Divalent ions are known to increase both the activity and stability of *E. coli* complex I (*Sazanov et al., 2003*). One of the calcium ligands, [G]Asp731, is part of

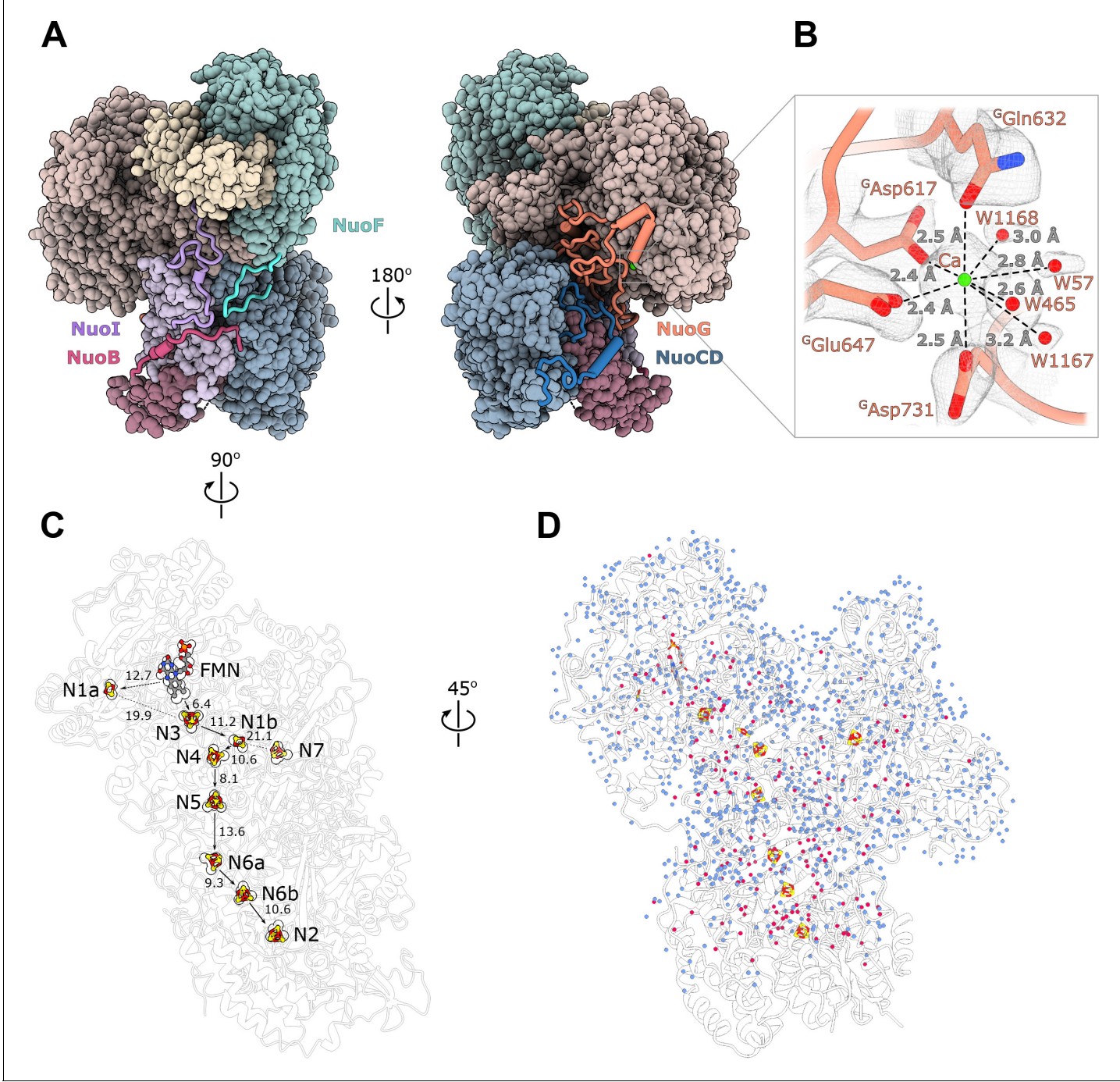

**Figure 2.** Structure of the peripheral arm. (**A**) The *Escherichia coli* - specific extensions in the peripheral arm subunits: C-termini of NuoI (violet), NuoB (pink), NuoF (turquoise), NuoG insertion (orange), and NuoCD linker (blue). (**B**) Structural details of the calcium-binding site. (**C**) Comparison of the FMN and Fe-S clusters positions in *E. coli* (shown as atoms) and *Thermus thermophilus* (shown as outline around *E. coli* atoms). Edge-to-edge distances and the electron pathway are indicated. (**D**) Water molecules modeled into the 2.1 Å resolution density map of the peripheral arm are show in blue. Water molecules conserved with the peripheral arm of ovine complex I (PDB ID: 6ZK9, red) are shown as red spheres. FMN and iron-sulfur (Fe-S) clusters are shown as spheres.

The online version of this article includes the following figure supplement(s) for figure 2:

**Figure supplement 1.** Conservation of *E. coli*-specific tails in the peripheral arm subunits.

**Table 3.** Properties of *E. coli* peripheral arm extensions (analyzed in PIZA).

| Subunit | Extension residues numbers | Interacts with subunit | Interaction surface [Å²] | Secondary structure | Specific interactions | Spatial overlap with subunits in other species |
|---|---|---|---|---|---|---|
| NuoF | C-term 424–445 | NuoI, NuoD | 254 | no | Nb 8 Sb 0 | Nqo15 *T. Therophilus* |
| | | NuoB | 557 | | Hb 0 Sb 1 | NUIM,NUZM,**NUMM***, *Y. Lipolytica* |
| | | | 118 | | Hb 0 Sb 0 | **Ndufs6**, Ndufs8, mouse |
| NouG | Insertion 687–781 | NuoCD | 1080 | two helical turns | Hb 12 Sb 5 | Nqo5 *T. Therophilus* |
| | | NuoI | 585 | | Hb 13 Sb 4 | NUGM, NUYM *Y. Lipolytica* |
| | | | | | | Ndufs3, Ndufs4 mouse |
| NuoI | C-term 139–180 | NuoG | 805 | one helical turn | Nb 8 Sb 5 | Nqo15 *T. Therophilus* |
| | | NuoB | 348 | | Nb 1 Sb 0 | **NUMM** *Y. Lipolytica* |
| | | NuoF | 292 | | Nb 3 Sb 0 | **Ndufs6** mouse |
| | | NuoD | 122 | | Nb 0 Sb 1 | |
| | | NuoE | 421 | | Nb 1 Sb 0 | |
| NuoB | C-term 196–220 | NuoI | 1095 | two helical turns | Nb 16 Sb 4 | NUIM, N7BM *Y. Lipolytica* |
| | | NuoD | 216 | | Nb 1 Sb 3 | Ndufs8, Ndufa12 mouse |
| | | NuoF | 117 | | Ng 0 Sb 0 | |

*Subunits in bold are homologs of NuoI.

the G-loop, suggesting that Ca²⁺ stabilizes the fold of the G-loop and consequently, the peripheral arm.

The extensions spatially overlap with the supernumerary subunits of complex I from *T. thermophilus* (*Sazanov and Hinchliffe, 2006*) and the structurally conserved supernumerary subunits of eukaryotic complex I (*Zhu et al., 2016*; *Table 3*), consistent with the suggestion that the primary role of supernumerary subunits is to stabilize the complex (*Fiedorczuk et al., 2016*).

## Bound water molecules

At 2.1 Å resolution, 1170 water molecules associated with the peripheral arm were modeled (*Figure 2D*). The positions of 180 water molecules are conserved with those identified in the peripheral arm of ovine complex I (*Kampjut and Sazanov, 2020*; *Figure 2D*, red spheres). Most of conserved waters are buried in the interior of the subunits, shielded from the solvent, and likely play a structural role in stabilizing the subunit fold. Only a few waters interact closely with iron-sulfur

**Table 4.** Comparison of hydrogen bond networks surrounding the N1a cluster in complex I structures solved at high resolution.

| Cluster | Hb acceptor | Hb donor | | |
|---|---|---|---|---|
| | | *E. coli* this work | *A. aeolicus* [pdb: 6hla] | *O. aries* [pdb: 6zk9] |
| N1a | N1a S1 | NH Asn136 3.5 Å | NH Ala130 3.6 Å | NH Ala147 3.3 Å |
| | | NH Leu134 3.1 Å | NH Leu128 3.5 Å | NH Leu145 3.3 Å |
| | | NδH Asn142 3.6 Å | | |
| | N1a S2 | NH CyS87 4.0 Å | NH Cys91 3.5 Å | NH Cys108 3.7 Å |
| | | | | OγH Thr105 2.4 Å |
| | Sγ, Cys92 (86,103)* | NH Ser94 3.5 Å | NH Ser88 3.5 Å | NH Thr 105 3.7 Å |
| | Sγ, Cys97 (91,108) | Nδ2H Asn142 4.1 Å | NH Val136 3.4 Å | NH Met153 4.3 Å |
| | | NH Asn 142 3.7 Å | | |
| | | OH W74 3.2 Å | | |
| | Sγ, Cys133 (127,144) | OH W127 3.3 Å | OH W794 3.2 Å | OH W649 3.2 Å |
| | | NH Gly97ᶠ4.0 Å | N GLy99ᶠ3.7 Å | NH GLy103ᶠ4.5 Å |
| | | NH Gly135 3.3 Å | N Gly129 3.5 Å | NH Gly146 3.3 Å |
| | Sγ, Cys137 (131,148) | N GLy97F 3.4 Å | N Gly99F 3.3 Å | N Gly103 3.2 Å |

*Numbering in parenthesis is given for *A. aeolicus* and *O. aries*, respectively.

clusters and may influence their potential (*Tables 4* and *5*). The water molecules close to or between iron-sulfur clusters are not more conserved than those in the other parts of the complex, suggesting that they were not evolutionary selected to optimize the rate of electron transfer as was suggested (*Schulte et al., 2019*).

At 2.1 Å resolution, several unusual density features were observed next to some surface-exposed histidines and between some cysteine-methionine pairs as listed in *Table 6* and depicted in *Figure 3—figure supplement 1*.

## Electron input and output sites

The FMN conformation and key water molecules in the NADH-binding pocket are conserved (*Kampjut and Sazanov, 2020*; *Schulte et al., 2019*). This includes W1060, hydrogen bonded to the isoalloxazine ring N5 atom of FMN and to $^F$Glu92, which likely acts as the activating group during catalysis of hydride transfer from NADH (*Fraaije et al., 2000*; *Figure 3A*). Schulte et.al. (*Schulte et al., 2019*) suggested a mechanism for regulation of reactive oxygen species (ROS) generation by *E. coli* complex I that involves flipping the carbonyl oxygen of $^F$Glu93 upon enzymatic reduction. Our structure unambiguously places the corresponding carbonyl oxygen in a

**Table 5.** Differences in the hydrogen bond network of iron-sulfur clusters in complex I structures solved at high resolution and water molecules in the immediate cluster environment.
(Only the clusters for which such comparison could have been done and clusters displaying differences in the environment are listed).

| | Organism | | |
|---|---|---|---|
| **Cluster, Subunit** | ***E. coli* this work** | ***A. aeolicus* [pdb: 6hla]** | ***O. aries* [pdb: 6zk9]** |
| N3, NuoF | His400 (+) | Leu395(-) | Leu407(-) |
| | Trp363(-) | Glu349(+) | Gln361(+) |
| | Asn196(+) | His198(-) | Lys202(-) |
| N1b, NuoG | HOH386 | | HOH1070 |
| N7, NuoG | HOH441 | | |
| | HOH577 | | |
| | Cys228 | | Asp229 |
| | Cys231 | | Asp232 |
| | Cys235 | | Ser236 |
| | Cys263 | | Ser264 |
| | | | HOH929* |
| | | | HOH933* |
| | | | HOH1005* |
| N4, NuoG | Thr203(+) | | Val205(-) |
| N5, NuoG | conserved | | |
| N6a, NuoI | Phe92(-) | | Tyr109(+) |
| | | | HOH539 |
| N6b, NuoI | Leu48(-) | | His65(+) |
| | Cys74(+) | | Ala91(-) |
| | Leu116(-) | | Glu133(+) |
| N2, NouB | HOH438 | | HOH353 |
| | HOH211 | | HOH579 |
| | Arg250$^D$ | | Arg85 dimethylated |
| | Ser62 | | Ala53 |

*Water molecules replacing the N7 cluster.

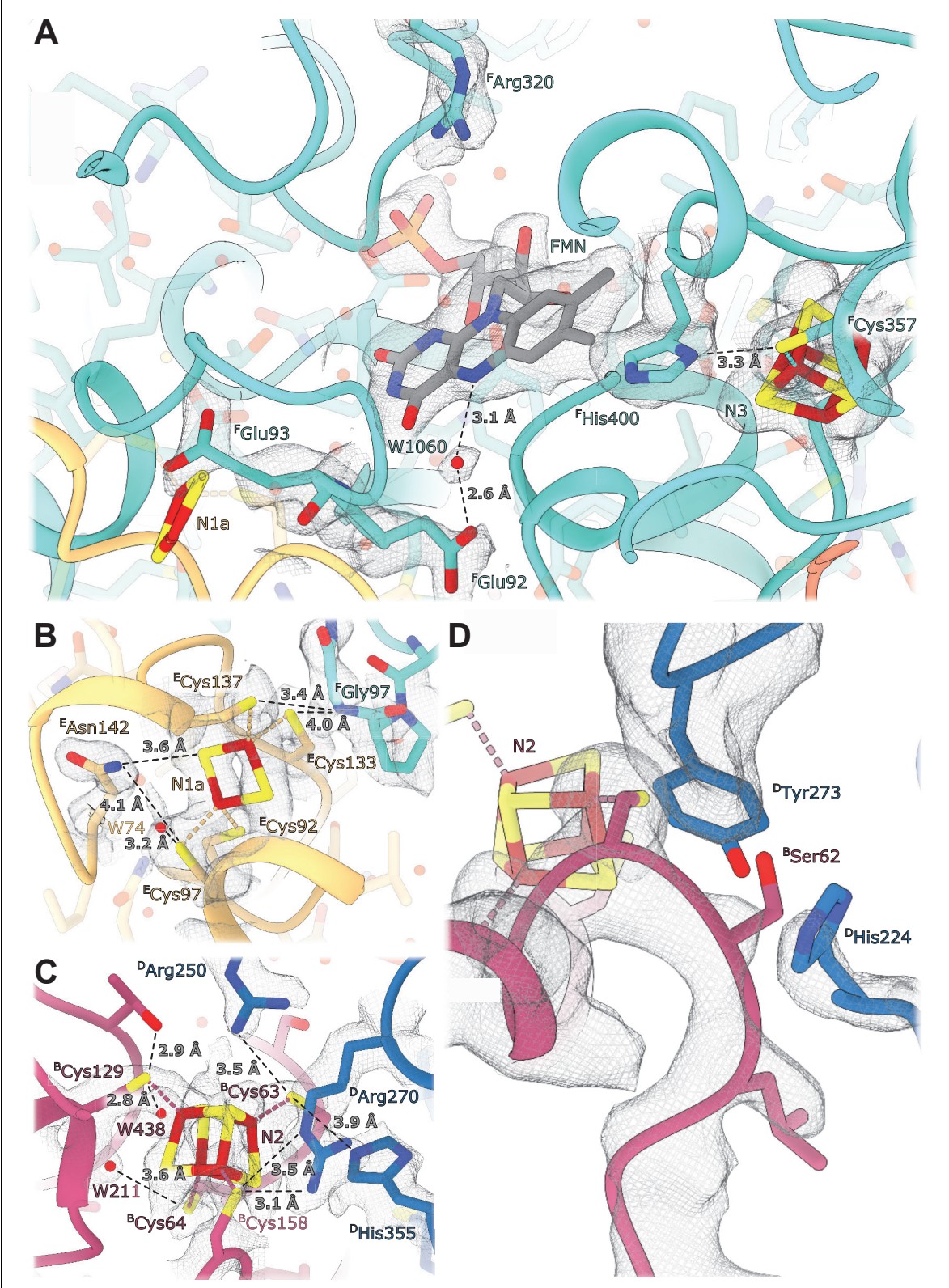

**Figure 3.** Details of the electron transport chain. (**A**) The NADH-binding pocket and environment of the Fe-S cluster N3. (**B, C**) Environment of the Fe-S clusters N1a and N2. (**D**) The ubiquinone binding-site.

The online version of this article includes the following figure supplement(s) for figure 3:

*Figure 3 continued on next page*

*Figure 3 continued*

**Figure supplement 1.** Unusual density features.

conformation that points away from FMN (*Figure 3A*) similar to conformations found in the reduced and oxidized ovine complex I (*Kampjut and Sazanov, 2020*), which does not support its involvement in ROS regulation.

*E. coli*-specific features in the FMN-binding pocket include [F]His400 that replaces the Leu residues found in other homologues. [F]His400 is in Van der Waals contact with the isoalloxazine ring of FMN; its imidazole ring interacts directly with the N3 cluster iron atom and forms a hydrogen bond with $S\gamma$ of [F]Cys357 coordinating N3 (*Figure 3A*). [F]His400 is solvent-accessible even in the presence of NADH, and therefore, may become protonated upon N3 reduction. [F]Arg320 is positioned such that it can form hydrogen bonds with the ribose moiety of the NADH nicotinamide group and may stabilize bound dinucleotide (*Figure 3A*). Both Arg320[F] and His400[F] may serve to counter-balance the negative charges of electrons on N1a and N3 clusters and to increase protein stability. The structure does not reveal specific features explaining the decreased affinity for FMN in the reduced enzyme (*Holt et al., 2016*). This can be attributed to minor conformational changes in the pocket upon enzyme reduction.

The Q-binding site in complex I is formed at the end of a crevice between NuoD and NuoB subunits (*Baradaran et al., 2013*). In *E. coli,* this wedge is formed by the 58–69 stretch of NuoB and the tip of the 220–225 loop from subunit NuoD. Both [D]Tyr273 and [D]His224, found in the proximity of bound decylubiquinone (*Baradaran et al., 2013*) are conserved in *E. coli* and point toward the quinone binding site (*Figure 3D*), whereas the tip of the 218–223 loop is flexible as in many complex I structures.

## Environment and potentials of iron-sulfur clusters

At a resolution of 2.1 Å the atoms constituting the iron-sulfur clusters are resolved as independent density blobs. The conformation of side chains as well as the positions of hydrating waters in the primary and secondary interaction spheres are mostly unambiguously resolved (*Figure 3*). In *E. coli* complex I, cluster N1a can be reduced by NADH due to its uniquely high potential ($\sim -0.3$ V), differentiating it from other characterized species in which N1a cannot be reduced by NADH (*Birrell et al., 2013*; *Zu et al., 2002*). The potential of iron-sulfur clusters in proteins among other factors depends on solvent exposure, proximity of charged residues, and the number of hydrogen bonds formed between the cluster environment and sulfur atoms of clusters and coordinating cysteines (*Denke et al., 1998*; *Fritz et al., 2002*). Comparison of the chemical environment of N1a with other high-resolution structures of complex I revealed three specific differences explaining the higher potential of the N1a cluster (*Table 4*): (1) *E. coli*-specific [E]Asn142 forms a hydrogen bond with $S\gamma$ of [E]Cys97 coordinating the N1a cluster and with N1a S1 (*Figure 3B*), consistent with its mutation to Met decreasing potential by 53 mV (*Birrell et al., 2013*). (2) In *E. coli*, water molecule

**Table 6.** Histidine residues with unassigned features extending from the imidazole ring.

| Residue | Modeled atom | Comments |
| --- | --- | --- |
| [E]His87 | None | Interaction with [E]D146 |
| [E]His150 | HOH | Distance 2.1 Å |
| [E]His152 | HOH | Distance 2.4–2.6 Å |
| [G]His5 | HOH | Density on both sides |
| [G]His101 | HOH | Positive environment |
| [G]His123 | HOH | Distance 2.2 Å |
| [G]His427 | HOH | Distance 2.85 Å |
| [G]His653 | HOH | Distance 2.5 Å, very strong |
| [CD]His163 | HOH | Distance 2.5 Å |
| [CD]His507 | HOH | Distance 4.2 Å |

W74 forms a hydrogen bond with Sγ of [E]Cys97. This water resides in a hydrophilic cavity created by *E. coli* specific [E]Gly140, replacing the alanine residue found in other species. (3) Because of small differences in the backbone conformation of NuoF, the backbone nitrogen of [F]Gly97 forms a hydrogen bond with Sγ of [E]Cys133 in *E. coli* and *Aquifex aeolicus* but not in *Ovis aries* (*Figure 3B*, *Table 4*).

The environment of the other iron-sulfur clusters is mainly conserved. The differences in hydrogen donors to the clusters, cysteine sulfur atoms, and water molecules in the cluster vicinity are listed in *Table 5* Clusters N3 and N2 are briefly discussed below as being the most interesting.

Cluster N3 interacts with *E. coli*-specific [F]His400 (*Figure 3A*); however, the potential of N3 is very similar between the species (*Leif et al., 1995*; *Yagi and Matsuno-Yagi, 2003*). The effect of proximal His residue is likely compensated by [F]Trp363 replacing the hydrogen bond donors (Glu or Gln) found in other species (*Table 5*).

The potential of cluster N2, the electron donor to quinone, varies in different species (*Hirst and Roessler, 2016*) notably being lower in *E. coli* compared to its mammalian analogues (−220 mV *vs.* −140 mV, respectively). However, the structure shows that the polar environment of N2 is very conserved (*Figure 3C*), including two water molecules, W211 and W438. Two arginines found in close proximity to the N2 cluster, Arg270[D] and Arg250[D], have conserved positions despite [49kDa]Arg85 in the mammalian homologue ([D]Arg250) being dimethylated (*Carroll et al., 2013*). This modification prevents it from forming a hydrogen bond with [B]Cys63, which should decrease N2 potential in the mitochondrial enzyme. Therefore, computational modeling, now enabled by the high-resolution structure, will be required to explain the difference in the cluster potentials.

## Structure of the membrane arm

The model of complete membrane arm, including the previously missing subunit NuoH (*Efremov and Sazanov, 2011*), was built into the density map with local resolution better than 3.5 Å at the arm center and approximately 4.0 Å at its periphery (*Figure 1A*, *Figure 1—figure supplement 4*). An additional density belt corresponding to the lipid nanodisc is clearly visible around the membrane-embedded region (*Figure 1A*, *Video 1*). It is flat in the plane of the membrane with a thickness of approximately 30 Å, and closely matches hydrophobic surface of the membrane arm. The belt locally bends next to the subunit NuoL at the region where it interacts with the long amphipathic helix and amphipathic helix connecting [H]TMH1-TMH2 ([H]AH1) protrudes into the nanodisc (*Video 1*).

The structure of the membrane arm in the lipid nanodisc is very similar to the crystal structure of the detergent-solubilized membrane arm (*Efremov and Sazanov, 2011*) [RMSD of 1.1 Å (1888 Cα)] (*Figure 4—figure supplement 1*). The curvature of the membrane arm observed previously (*Efremov and Sazanov, 2011*) was unchanged in the lipid environment, and therefore, is not an artifact of crystallization or solubilization (*Verkhovskaya and Bloch, 2013*). Local structural differences in crystal structure include expected repositioning of [A]TMH1 next to [H]TMH2 (*Baradaran et al., 2013*), and a change in conformation of the [M]TMH5-TMH6 loop (*Figure 4—figure supplement 1*).

The fold of subunit NuoH is similar to the structures of *T. thermophilus* and of eukaryotic complexes with one important exception. The density for the N-terminus of NuoH (residues 1–52) that includes [H]TMH1 and a part of [H]TMH1-TMH2 loop, is completely missing in the reconstructions of the membrane fragment and of the complete complex (*Figure 1*, *Video 1*).

The structures visualize a complete chain of charged residues connecting the Q-site with charged residues in antiporter-like subunits (*Figure 4A*). We analyzed the environment of ionizable residues found within the 'E-channel' (*Baradaran et al., 2013*), a region situated between the Q-cavity and antiporter-like subunit NuoN, to evaluate the presence of a continuous proton translocation path linking the Q-cavity with the antiporter-like subunits suggested for ovine complex I (*Kampjut and Sazanov, 2020*).

The trans-membrane region of *E. coli* NuoH contains fewer charged residues than its structurally characterized homologs (*Figure 4—figure supplement 2*). Here, only *E. coli*-specific [H]His208, separated from [H]Glu157 by 12 Å, is found in the center of NuoH (*Figure 4A*). A large hydrophilic cavity stretches from the Q-site towards the center of subunit NuoH and ends next to the invariant **[H]Glu157** (hereafter, invariant residues are marked in bold), suggesting that this glutamic acid can exchange protons with a bound ubiquinone.

The region between NuoH and NuoN includes six ionizable side chains located in the middle of the membrane bilayer, four of which are invariant (*Figure 4A*). The distances between the residues

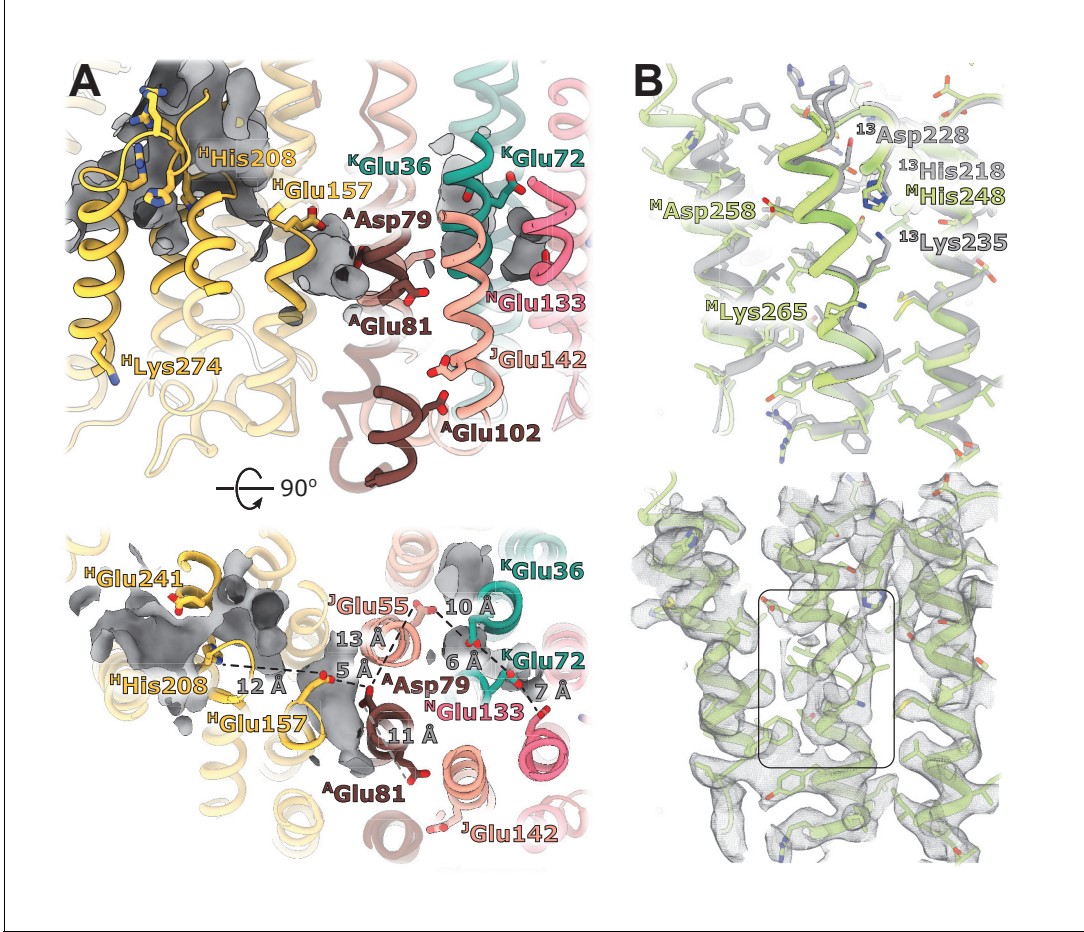

**Figure 4.** Structural details of the membrane arm. (**A**) The E-channel. Top: Side view, bottom: view from the cytoplasm. Charged residues between NuoH and NuoN subunits are indicated along with the distances between them. The cavities allowing entrance of ions and water molecules are shown as gray surfaces. (**B**) Conformational heterogeneity within the NuoM subunit. Top: Comparison of the conformation of [M]TMH8 in *E. coli* (green) with [13]TMH8 in *T. thermophilus* (gray). Bottom: density of NuoM subunit displays heterogeneity within [M]TMH8 region, marked by a black rectangle. The online version of this article includes the following figure supplement(s) for figure 4:

**Figure supplement 1.** Comparison of X-ray and cryo-EM structures of the membrane domain.

**Figure supplement 2.** Conserved chain of ionizable residues.

vary from 5 Å to 12 Å which requires either displacement of the side chains or bridging water molecules to facilitate proton exchange between them. Analysis of cavities and potential hydration sites using DOWSER++ (*Morozenko and Stuchebrukhov, 2016*) shows that the residues [H]**Glu157** and [A]**Asp79** along with the carbonyl oxygen of [J]Gly61 on the π-bulge of [J]TM3 (*Efremov and Sazanov, 2011*) point into a hydrophilic cavity that can accommodate several water molecules enabling proton exchange between the ionizable residues. Similarly, cavities that can be hydrated link a chain of ionizable residues [J]Glu55-[K]**Glu36**-[K]Glu72-[N]Glu133 potentially enabling proton exchange between its ends. The residues [A]**Asp79** and [J]Glu55/ [K]**Glu36** are separated by a distance exceeding 12 Å and a region packed with hydrophobic residues, making proton exchange between the Q-site and NuoN unlikely. [A]**Glu81**, located opposite [A]**Asp79** on [A]TMH2, apparently does not participate in linking [A]**Asp79** with [N]**Glu133**. However, it faces hydrophilic environment of [J]Ser145, *E. coli*-specific [J]Glu142, and [A]Glu102, potentially linking it to the periplasmic surface (*Figure 4A*). Interestingly, mutation of any individual carboxylic groups in the [A]**Glu81**/[A]**Asp79** pair does not affect pumping, while mutation of both residues completely abolishes the protein activity (*Kao et al., 2004*), suggesting that they are functionally important but interchangeable. Thus, analysis of the protein translocation pathways indicates that in *E. coli,* no continuous proton path exists between the Q-site and NuoN.

Curiously, [H]Lys274, almost universally conserved in complex I and related hydrogenases, is found in the [H]TMH7 off the main pathways proposed for proton translocation (*Figure 4A*), however, the length and flexibility of the side chain allow it to switch between the extracellular surface and center of the membrane.

The cytoplasmic half of [J]TMH3 found to assume two alternative conformations in eukaryotic complex I (*Agip et al., 2018*; *Kampjut and Sazanov, 2020*) is very well-resolved in our reconstructions, suggesting the absence of alternative conformations in *E. coli* complex I. A peculiar feature is observed in subunit NuoM instead.

The density of the cytoplasmic half of [M]TM8 is poor and fragmented between residues 255 and 265, indicating the existence of multiple conformations (*Figure 4B*, *Video 2*). This region is buried in the middle of NuoM and the density of surrounding helices is very well-resolved indicating local character of the disorder. This region spans the invariant [M]**Lys265,** includes the π-bulge, and [M]Asp258 in some bacteria. Interestingly, in *T. thermophilus* the cytoplasmic region of [13]TM8 is rotated by two residues relative to *E. coli* structure (*Figure 4B*) which results in repositioning of *T. thermophilus* [13]**Lys235** ([M]**Lys265** in *E. coli*) from the center of the second structural repeat (TM9-TM13) towards the interface between the structural repeats facing [13]**His218** ([M]**His248** in *E. coli*). Thus, higher mobility of the helical fragment situated at a critical position at the interface of symmetry-related modules may indicate π-bulge-enabled helical rotation and its potential role in proton translocation.

## The peripheral-membrane arm interface

The interface between membrane and peripheral arms mediates the coupling of ubiquinone reduction to proton translocation across the membrane. It is highly conserved between complex I homologs and related membrane-bound hydrogenases (*Baradaran et al., 2013*; *Grba and Hirst, 2020*; *Kampjut and Sazanov, 2020*; *Yu et al., 2020*; *Yu et al., 2018*; *Figure 5—figure supplement 1*) and forms through interaction between subunits NuoB and NuoD of the peripheral arm with NuoH and the [A]TMH1-TMH2 loop of the membrane domain.

Apart from several interfacial regions of subunits NuoB, NuoD, and NuoI that become more ordered upon complex formation (*Figure 5—figure supplement 2*, *Table 2*), the membrane-facing surface of *E. coli* peripheral arm, including the residues lining the Q-cavity, is highly mobile in both dissociated and complexed arms (*Figure 1—figure supplement 4*, *Figure 5—figure supplement 2*). This suggest that the interfacial region of the peripheral arm is inherently flexible and likely responsible, at least in part, for the high relative mobility of the arms. Similar to complex I from *T. thermophilus* (*Baradaran et al., 2013*), there are no specific conformational changes at the interface upon association of the arms.

Structure of the NuoH surface and relative arrangement of subunits NuoB and NuoD in *E. coli* complex I is similar to that of complex I from other species, however, their relative positions differ. In *E. coli* complex I, NuoB and NuoD are rotated around an axis passing through the center of NuoH and the interface between NuoF and NuoG by approximately 15 degrees anticlockwise when observed from the top of the peripheral arm (*Figure 5A*). This results in over 10 Å shift of four-helical interfacial region of NuoD, and its separation from NuoH (*Figure 5B*). The highly conserved fragment of the [A]TMH1-TMH2 loop (residues 46–53), that plugs the crevice between NuoD and NuoB (*Figure 5—figure supplement 1B*) and interacts with the ubiquinone-coordinating loop [D]221-228, is also disordered (*Figure 5B*).

On the opposite side of the interface, structural rearrangements include a 7-degree tilt of [A]TMH1 that becomes more perpendicular to the membrane plane and approximately 15-degree rotation of [H]AH1 in the direction of [H]TMH1 and towards the membrane center (*Figure 5C*)

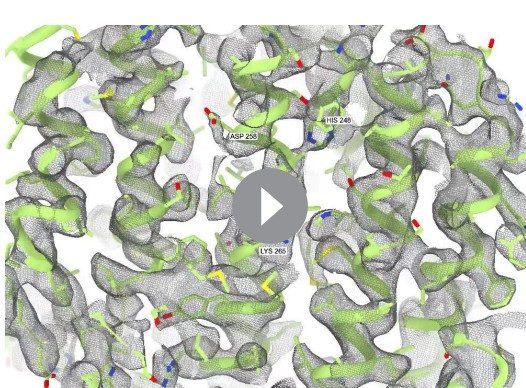

**Video 2.** The fragmented density of the NuoM TM8 is surrounded by well-resolved TMHs.
https://elifesciences.org/articles/68710#video2

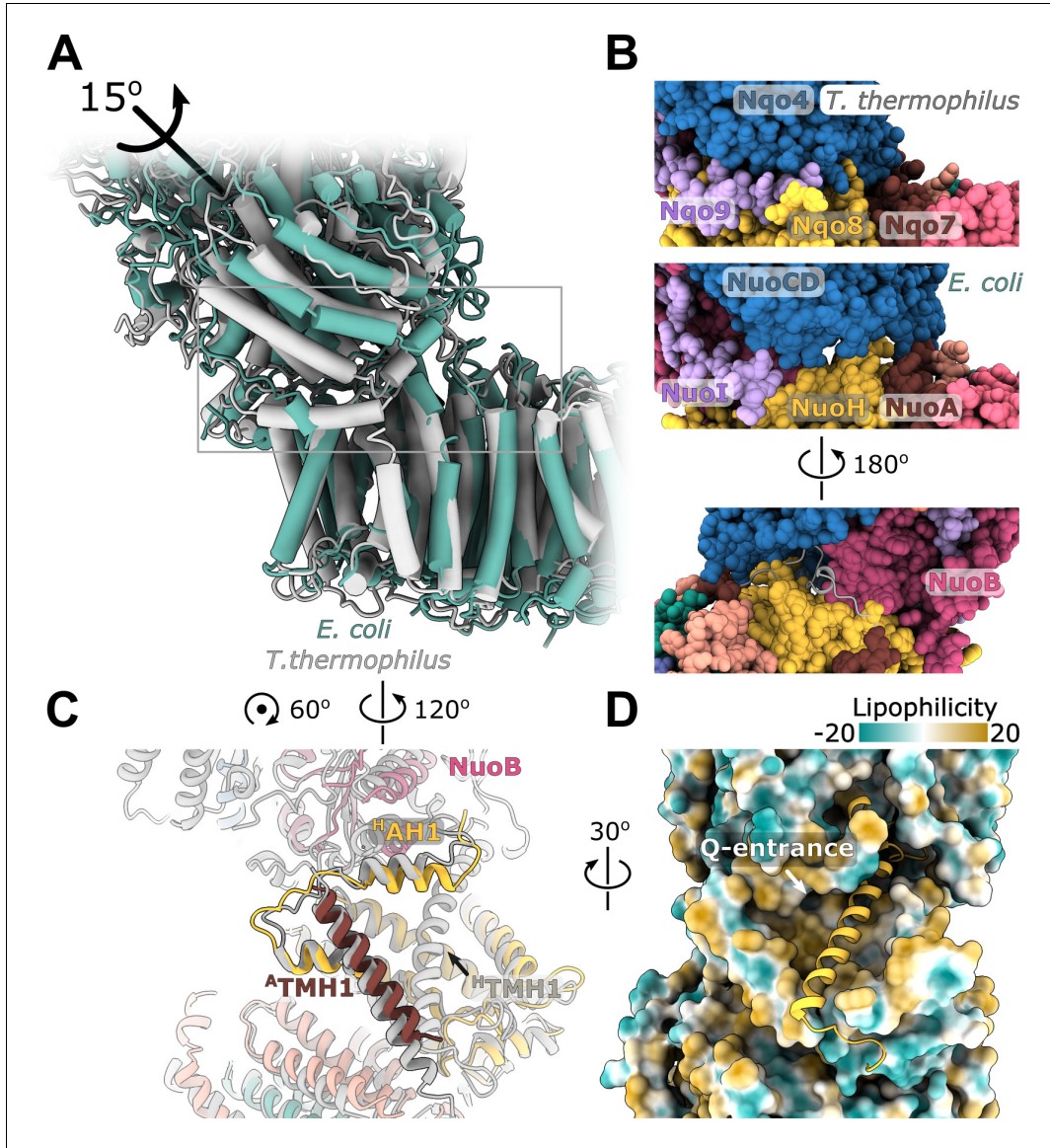

**Figure 5.** Interface between the peripheral and membrane arms. (**A**) Comparison of the interface between *E. coli* (green) and *T. thermophilus* (PDB ID:4HEA, gray) complex I. Structures were aligned on the subunit NuoH/Nqo8. The rotation axis of the subunits NuoB/D module relative to Nqo6/4 is indicated. (**B**) Interfacial contacts between the peripheral and membrane arms in *T. thermophilus* (upper panel) and *E. coli* (middle and bottom panel). A gap in the subunit interface is apparent in the absence of the conserved ᴬTMH1-2 loop fragment. The corresponding loop from *T. thermophilus* is shown in gray in the cartoon representation (bottom panel). (**C**) Differences in the structures of NuoH and NuoA subunits between *E. coli* (color coded as in *Figure 1*) and *T. thermophilus* (gray). (**D**) View from the membrane on the entrance to the Q-cavity. Homology model of ᴴTM1, absent in the *E. coli* structure, is shown in the cartoon representation. The protein surface is colored by lipophilicity.

The online version of this article includes the following figure supplement(s) for figure 5:

**Figure supplement 1.** Conserved interface between the arms.

**Figure supplement 2.** B-factors of the peripheral arm show higher mobility at the arms interface.

reducing opening of the Q-cavity entrance (*Figure 5D*). Homology modeling indicates that the observed rearrangements are still compatible with ᴴTMH1 occupying its expected position without any steric clashes (*Figure 5D*).

Rotation of the NuoB/NuoD module creates multiple openings in the interface between the arms (*Figure 5B*). They are large enough to allow water and proton exchange between the Q-cavity and

bulk solvent, suggesting that the ubiquinone bound within the Q-site can receive protons directly from the solvent. Therefore, we think the resolved conformations of complex I represent uncoupled states in which redox reaction is not coupled to proton translocation.

To better understand the reasons for the observed uncoupled conformation and the missing density for ᴴTMH1, we purified *E. coli* complex I in detergent LMNG, showed that it can catalyze redox reactions (*Figure 6—figure supplement 1*) and solved its structure to resolution of 6.7 Å (*Figure 6—figure supplement 2*). The detergent-solubilized complex also displays high relative mobility of the arms (*Figure 6—figure supplement 3*) and has uncoupled conformation (*Figure 6*). Its peripheral arm is rotated even further away from the expected coupled state position than in the nanodisc-reconstituted structures. Both the cryo-EM sample preparation conditions and more homogeneous distribution of particle orientations indicate that interaction of the complex with air-water interface was significantly reduced when compared with the complex in nanodiscs. This allows us to conclude that neither air-water interface nor reconstitution into nanodiscs cause the uncoupled conformations.

The ᴴTMH1 helix is resolved in the detergent-solubilized complex (*Figure 6A*). Its density is weaker than that of the surrounding helices and it is strongly bent (*Figure 6B*). Simultaneously, ᴴAH1 takes the conformation resembling other complex I homologs while ᴬTMH1 bends towards the arm core. The arrangement of helices in detergent-solubilized reconstruction appears to be more compact and more bent than in the lipid environment which may restrain the otherwise more flexible ᴴTMH1.

## Discussion

### Structural features of *E. coli* complex I

*E. coli* complex I is composed of the smallest number of subunits among all structurally characterized complex I homologs. Yet, it still evolved a strategy to stabilize peripheral arm assembly without involving additional subunits. The interactions between subunits are stabilized by extended C-termini and a large G-loop (*Figure 2*), which is further stabilized by the $Ca^{2+}$ ion known to modulate

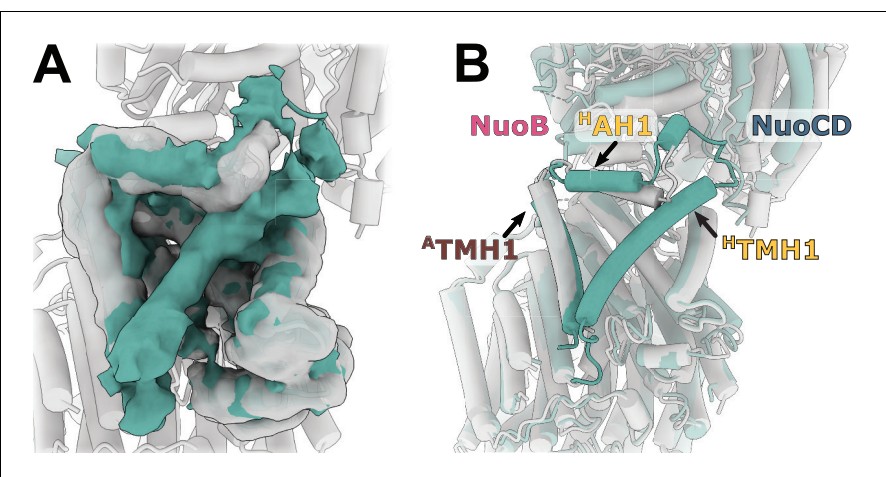

**Figure 6.** Reconstruction of *E. coli* complex I solubilized in detergent reveals uncoupled conformation with resolved ᴴTMH1. (**A**) Comparison of the density in the ᴴTMH1 region between the focused reconstruction of membrane arm in lipid nanodisc filtered to 7 Å (gray) and LMNG-solubilized entire complex map (green). (**B**) Overlay of the conformation 1 solved in nanodiscs (gray) and the model fitted into the LMNG-solubilized reconstruction.

The online version of this article includes the following figure supplement(s) for figure 6:

**Figure supplement 1.** Purification and biochemical characterization of *E. coli* complex I in LMNG.

**Figure supplement 2.** Processing of LMNG-solubilized cryo-EM data.

**Figure supplement 3.** Dynamic connection between peripheral and membrane arms in the LMNG – solubilized complex I.

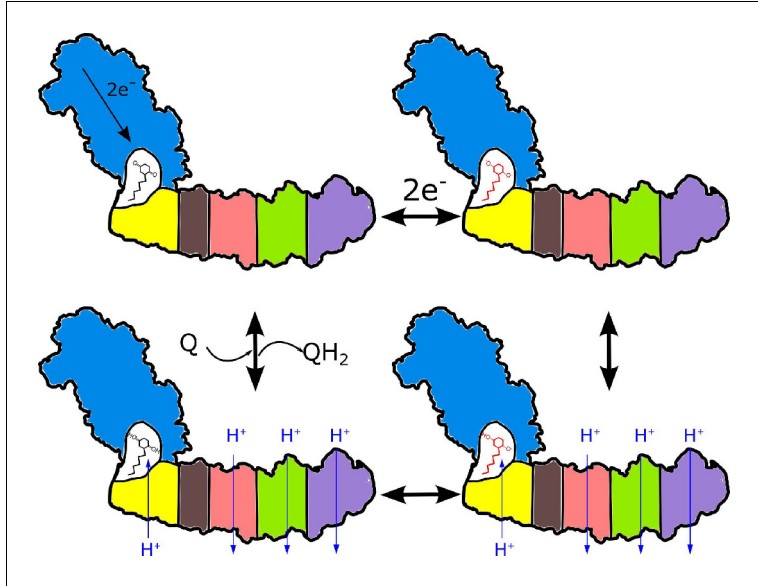

**Figure 7.** Proposed coupling mechanism in respiratory complex I. Ubiquinone reduction decreases the proton potential in the Q-cavity, generating/enhancing potential difference between the Q-cavity and periplasmic space. It is subsequently neutralized by protons translocated through NuoH from the periplasm into the Q-cavity and this translocation is coupled with the reversible translocation of three protons into the periplasm. Color coding of the schematic subunits in the membrane arm is similar to that described in *Figure 1*, negatively charged states of ubiquinone are shown in red.

the complex stability (*Sazanov et al., 2003*). This indicates existence of evolutionary pressure on maintaining the peripheral arm integrity, which was 'solved' in a species-dependent manner.

The membrane arm structure reveals no apparent continuous proton translocation path between the Q-cavity and subunit NuoN. Further, there are no indications for the existence of different conformations in the cytoplasmic half of ^JTMH3 observed in mammalian complex I (*Agip et al., 2018*; *Kampjut and Sazanov, 2020*) attributed to deactive-active transition (*Agip et al., 2018*) or more recently, to different catalytic intermediates (*Kampjut and Sazanov, 2020*). This suggests that these states are either suppressed in the resting state of the bacterial complex or do not occur at all. Conversely, ^MTMH8 displays localized disorder next to the π-bulge, indicating possible involvement of this helix in the structural rearrangements associated with proton translocation, and to our knowledge, this represents the first indication of specific conformational changes in antiporter-like subunits.

Purified *E. coli* complex I is known to be more flexible and fragile than its homologs from other organisms (*Morgan and Sazanov, 2008*; *Sazanov et al., 2003*). Our cryo-EM reconstructions reveal the reasons for its high flexibility. The peripheral and membrane arms are mainly rigid, whereas the connection between arms is flexible (*Figure 1C*, *Figure 1—figure supplement 5*). The high mobility of the interfacial regions and the relative rotation of the arms disrupts conserved interfacial interactions and exposes Q-cavity to the solvent (*Figure 5A*). This differentiates *E. coli* complex I from its structurally characterized homologs in which the Q-cavity is sealed from the solvent. Thus, we interpret the observed conformation as an uncoupled state.

The reasons for such conformation are not clear. 3D cryo-EM reconstructions obtained for the detergent-solubilized and the nanodiscs-reconstituted complex I allow us to exclude the air-water interface or the nanodisc as the cause. In turn, the lipid bilayer mimetics provided by detergent micelle or lipid nanodiscs might not be sufficiently close to the native lipid bilayer, causing uncoupling. Alternatively, this conformation may have a functional origin and correspond to a resting state of *E. coli* complex I (*Belevich et al., 2017*) analogous to the deactive state in eukaryotic complex I (*Babot et al., 2014*). However, the latter have been associated with local conformational changes (*Agip et al., 2018*; *Parey et al., 2018*), rather than displacement of the complete domains.

The absence of $^H$TMH1 density in nanodiscs, but not in detergent, is another unique feature of *E. coli* complex I. $^H$TMH1 is exposed to the lipid environment and the width of the nanodisc next to $^H$TMH1 is similar to other regions around the membrane arm (*Video 1*). Moreover, homology modeled $^H$HTM1 fits the empty space without steric clashes suggesting that $^H$HTM1 is dynamic rather than displaced or unfolded. By comparing the detergent-solubilized and reconstituted complexes we can conclude that position and dynamics of this helix is neither the cause of the uncoupled conformation nor of the high relative mobility of the arms.

### Hypothetical coupling mechanism

The absence of a continuous proton-translocation pathway between the Q-site and subunit NuoN, as well as high flexibility of the peripheral arm interface are not consistent with the recently proposed coupling mechanisms relying on specific movements of the interfacial loops (*Cabrera-Orefice et al., 2018*; *Kampjut and Sazanov, 2020*). This led us to ask whether a coupling mechanism consistent with known complex I properties, but without the movements of interfacial loops is conceivable.

Imposing the microscopic reversibility constraints (*Astumian et al., 2016*; *Onsager and Machlup, 1953*) and the requirement for applicability of the mechanism to the entire class of evolutionarily related complexes we came up with a new, simple mechanism that to the best of our knowledge has not been considered previously. It is briefly outlined below with more details provided in Appendix.

Coupling can be enabled by the formation of a cavity isolated from external protons that is accessible to electron acceptors in their neutral form only. The requirement for a tightly coupled cavity is consistent with high conservation of the interface between the arms. Due to its small size, the redox potential of ubiquinone will be strongly modulated by the extraction/addition of single protons from/to the cavity (*Lemmer et al., 2011*), and reciprocally, activity of the protons in the cavity will be modulated by the changes of the ubiquinone redox state. Upon ubiquinone reduction, the decreased proton activity is rectified by proton transfer through NuoH from the extracellular compartment as shown in *Figure 7*. This proton transfer is coupled to the transfer of three protons through three antiporter-like subunits in the opposite direction resulting in a net transfer of 2H$^+$ per 1 e$^-$. Consequently, the entire membrane module of complex I functions as a proton antiporter with stoichiometry 1H$^+_{in}$/3H$^+_{out}$ and most likely operates by a classical alternating access mechanism (*Jardetzky, 1966*). The mechanism is applicable to evolutionary-related hydrogenases. It suggests that a pH jump creates a proton motive force between the sealed Q-cavity and periplasmic surface and might be used to trap equilibrium conformations associated with proton translocation by the membrane domain of complex I.

### Conclusions

Here we described the cryo-EM structures of *E. coli* respiratory complex I that reveal unique structural features. We discovered an evolutionary strategy specific to mesophilic bacteria for stabilizing the peripheral arm assembly through the extended C-termini, the G-loop, and the bound Ca$^{2+}$ ion. In the membrane arm, $^M$TMH8 displays dynamics unseen in the other complex I homologues which may reflect conformational flexibility associated with proton translocation. We also observed the relative rotation of the membrane and peripheral arms disrupting the conserved interface and trapping the complex in an uncoupled conformation. Whether this conformation is biologically relevant or is a result of protein purification is to be clarified by further research.

## Materials and methods

**Key resources table**

| Reagent type (species) or resource | Designation | Source or reference | Identifiers | Additional information |
|---|---|---|---|---|
| Gene (*Escherichia coli*) | *nuoe, nuof* | GenBank | NC_012971.2 region 2288438–2289174 | |
| Strain, strain background (*Escherichia coli*) | BL21-AI | Thermo Fisher Scientific Inc | C607003 | Chemically competent cells |

*Continued on next page*

*Continued*

| Reagent type (species) or resource | Designation | Source or reference | Identifiers | Additional information |
|---|---|---|---|---|
| Strain, strain background (*Escherichia coli*) | BL21FS | This study | | BL21-AI supplemented with a twin-STREP-tag and a TEV protease recognition site on the nuoF N-terminus, see Materials and Methods, "Generation of an *E. coli* strain…." |
| Recombinant DNA reagent | pCas (plasmid) | *Jiang et al., 2015* | RRID:Addgene_62225 | A vector for Crispr-Cas9 mutagensis |
| Recombinant DNA reagent | pTargetF (plasmid) | *Jiang et al., 2015* | RRID:Addgene_62226 | A vector for Crispr-Cas9 mutagensis |
| Recombinant DNA reagent | pMSP2N2 (plasmid) | *Grinkova et al., 2010* | RRID:Addgene_29520 | A vector for expression of MSP2N2 membrane scaffold protein |
| Recombinant DNA reagent | DNA knock-in cassette | This study | | NuoEF fragment supplemented with a twin-STREP-tag and a TEV protease recognition site, see Materials and Methods, "Generation of an *E. coli* strain…." |
| Chemical compound, drug | Piericidin A | Cayman Chemical | 15379 | |
| Chemical compound, drug | NADH | Carl-Roth GmbH | AE12.2 | |
| Chemical compound, drug | Decylubiquinone; DQ | Sigma Aldrich BVBA | D7911 | |
| Chemical compound, drug | Ubiquinone-1; Q1 | Sigma Aldrich BVBA | C7956 | |
| Chemical compound, drug | Potassium ferricyanide (III); FeCy | Sigma Aldrich BVBA | 244023 | |
| Chemical compound, drug | Lauryl Maltose Neopentyl Glycol; LMNG | Anatrace | NG310 | |
| Chemical compound, drug | n-Dodecyl-β-D-Maltopyranoside; DDM | Anatrace | D310 | |
| Chemical compound, drug | *E. coli* polar extract | Avanti Polar Lipids | 100600C | |
| Software, algorithm | SerialEM 3.0.8 | *Mastronarde, 2005* | RRID:SCR_017293 | |
| Software, algorithm | MotionCor2 | *Zheng et al., 2017* | RRID:SCR_016499 | |
| Software, algorithm | CTFFIND-4.1 | *Rohou and Grigorieff, 2015* | RRID:SCR_016732 | |
| Software, algorithm | crYOLO 1.4, crYOLO 1.7.0 | *Wagner et al., 2019* | RRID:SCR_018392 | |
| Software, algorithm | Relion 3.1 | *Zivanov et al., 2018* | RRID:SCR_016274 | |
| Software, algorithm | cryoSPARC 2.11, cryoSPARC 3.2.0 | *Punjani et al., 2017* | RRID:SCR_016501 | |

*Continued on next page*

*Continued*

| Reagent type (species) or resource | Designation | Source or reference | Identifiers | Additional information |
|---|---|---|---|---|
| Software, algorithm | SWISS-MODEL server | *Waterhouse et al., 2018* | RRID:SCR_018123 | |
| Software, algorithm | UCSF Chimera 1.13.1 | *Pettersen et al., 2004* | RRID:SCR_004097 | |
| Software, algorithm | Coot 0.9 | *Casañal et al., 2020* | RRID:SCR_014222 | |
| Software, algorithm | PHENIX 1.19.2 | *Liebschner et al., 2019* | RRID:SCR_014224 | |
| Software, algorithm | ISOLDE 1.0b5 | *Croll, 2018* | | |
| Software, algorithm | MolProbity | *Williams et al., 2018* | RRID:SCR_014226 | |
| Software, algorithm | ConSurf server | *Ashkenazy et al., 2016* | RRID:SCR_002320 | |
| Software, algorithm | UCSF ChimeraX 1.1 | *Goddard et al., 2018* | RRID:SCR_015872 | |
| Software, algorithm | The PyMOL Molecular Graphics System, Version 2.4.1 | Schrödinger, LLC | RRID:SCR_000305 | |
| Other | Quantifoil R0.6/1 Cu300 holey carbon grids | Quantifoil | Q350CR-06 | |

## Generation of an *E. coli* strain expressing Twin-Strep-tagged respiratory complex I

The native *nuo* operon encoding the 13 subunits of respiratory complex I (NuoA-N) was engineered with a Twin-Strep-tag (WSHPQFEKGGGSGGGSGGGSAWSHPQFEK, IBA GmbH) at the N-terminus of NuoF using CRISPR-Cas9-enabled recombineering (*Jiang et al., 2015*). The DNA sequence encoding the C-terminal region of NuoE and N-terminus of NuoF was retrieved from GenBank (Acc. No. NC_012971.2 region 2288438–2289174). The tag-coding sequence followed by a TEV protease recognition site (*Tropea et al., 2009*) was appended upstream of the NuoF N-terminus and was codon-optimized, together with the 2288766–2288807 region of the genomic fragment. Such designed, linear DNA knock-in cassette was synthesized (GenScript). The vectors pCas and pTargetF were gifts from Sheng Yang (Addgene plasmids #62225 and #62226). The N20 sequence (GGTCAGCGGA TGCGTTTCGG) was introduced into pTargetF by inverse PCR. Genomic engineering was performed according as described by *Jiang et al., 2015*. Briefly, pCas vector was transformed into the chemically competent *E. coli* BL21AI strain (Thermo Fisher Scientific Inc). The transformants were grown in shake-flask culture at 30°C in Lysogeny Broth (LB) medium containing 25 µg mL$^{-1}$ (w/v) kanamycin monosulfate and 10 mM L-arabinose. Upon reaching OD$_{600}$ 0.5, the bacteria were rendered electrocompetent and were co-electroporated with the linear DNA cassette and the mutated pTargetF vector. The transformants were selected on LB-agar plates supplemented with 25 µg mL$^{-1}$ (w/v) kanamycin and 50 µg mL$^{-1}$ (w/v) streptomycin, or 50 µg mL$^{-1}$ (w/v) spectinomycin. The positives, identified by colony PCR and DNA sequencing, were cured of the plasmids as described previously (*Jiang et al., 2015*). We further refer to the modified strain as *E. coli* BL21FS (NuoF-Strep).

## Expression and purification of respiratory complex I

*E. coli* BL21FS was cultivated in LB medium for 48 hr at 37°C in a microaerobic environment. The cells were harvested by centrifugation and the membrane fraction was isolated as described by *Sazanov et al., 2003*. All subsequent steps were performed at 4°C. The homogenate was solubilized in 2% (w/v) n-Dodecyl β-D-maltoside (DDM, Anatrace) for 2 hr while stirring, after which the non-solubilized fraction was removed by ultracentrifugation at 225,000 × *g* for 1 hr. The supernatant was adjusted to 200 mM NaCl and loaded on a 5 mL Strep-Tactin Superflow high capacity column (IBA GmbH). After washing with 25 column volumes (CV) of buffer A (50 mM Bis-tris pH 6, 2 mM CaCl$_2$, 200 mM NaCl, 0.04% [w/v] DDM, 10% [v/v] sucrose, 0.003% [w/v] *E. coli* polar lipid extract [Avanti

Polar Lipids, EPL], 0.2 mM PMSF), complex I was eluted with 2 CV of buffer B (buffer A containing 5 mM D-desthiobiotin [IBA GmbH]). The purity of the eluted protein was assessed by SDS-PAGE and activity assays (*Figure 1—figure supplement 2*). The purified complex I was concentrated using an Amicon Ultra-4 100K centrifugal filter (Merck) to 0.5 mg mL$^{-1}$ (w/v), fast-frozen in liquid nitrogen and stored at −80˚C.

For the preparation in lauryl maltose neopentyl glycol (LMNG), the protocol was modified as described below. Membranes were solubilized in 2% (w/v) LMNG (Anatrace). The buffer A-LMNG consisted of 50 mM Bis-tris pH 6, 2 mM CaCl$_2$, 200 mM NaCl, 0.03% (w/v) LMNG, 10% (v/v) sucrose, 0.2 mM PMSF. The buffer B-LMNG was the buffer A-LMNG supplemented with 5 mM D-desthiobiotin. The LMNG-purified complex I was concentrated using an Amicon Ultra-4 100K centrifugal filter (Merck) to 10 mg mL$^{-1}$ (w/v) and loaded on the Superose 6 Increase 10/300 GL column (GE Healthcare) equilibrated in 20 mM Bis-Tris pH 6.0, 200 mM NaCl, 2 mM CaCl$_2$ and 0.003% (w/v) LMNG. The protein-containing fractions were pooled, concentrated to 2–3 mg mL$^{-1}$ (w/v) using Amicon Ultra-0.5 100K centrifugal concentrators, and used for cryo-EM grid preparation.

## Reconstitution of respiratory complex I into lipid nanodiscs

The membrane scaffold protein MSP2N2 was expressed and purified following a published protocol (*Grinkova et al., 2010*). Purified, lipid-containing complex I preparation at 520 nM concentration was mixed with 10.4 µM MSP2N2 (1:20 protein:MSP molar ratio, no additional lipids were added during reconstitution) and incubated for 1 hr at 4˚C. Subsequently, the detergent was removed by adding 0.5 g mL$^{-1}$ (w/v) Bio-Beads (Bio-Rad) overnight at 4˚C. The reconstituted protein was further purified on the Superose 6 Increase 10/300 GL column (GE Healthcare) equilibrated in a buffer comprising 20 mM Bis-Tris pH 6.8, 200 mM NaCl and 2 mM CaCl$_2$. The protein-containing fractions were pooled and concentrated to 0.1–0.2 mg mL$^{-1}$ (w/v) using Amicon Ultra-0.5 100K centrifugal concentrators.

## Activity assays

NADH:ferricyanide (FeCy), NADH:ubiquinone-1 (Q1), and NADH:decylubiquinone (DQ) activities were measured as described previously (*Sazanov et al., 2003*). NADH:FeCy and NADH:Q1 activities of the nanodisc-reconstituted sample were assayed in the buffer containing 10 mM Bis-Tris pH 6.8, 200 mM NaCl, and 10 mM CaCl$_2$. To increase the solubility of DQ during NADH:DQ assays the assay buffer was supplemented with a small amount of LMNG (0.003%). All the assays for the LMNG-purified sample were performed in the LMNG activity buffer containing 10 mM Bis-Tris pH 6.0, 25 mM NaCl, 10 mM CaCl$_2$, and 0.1% LMNG.

For the NADH:FeCy assay, 0.9 nM of complex I and 1 mM FeCy (Sigma Aldrich BVBA) was added to the assay buffer in a stirred quartz cuvette. For the NADH:Q1/DQ, 3–9 nM of detergent-solubilized or nanodisc-reconstituted complex I and 100 µM Q1/DQ (Sigma Aldrich BVBA) was added to the assay buffer in a stirred quartz cuvette at 30˚C and incubated for 5 min. The reactions were initiated by adding 100 µM NADH (Carl-Roth GmbH) and followed as reduction in absorbance at 340 nm using a Varian Cary 300 UV-Vis spectrophotometer (Agilent Technologies, Inc). To perform the inhibition assay, 20 µM Piericidin A (Cayman Chemical) was added during the NADH:Q1 or NADH:DQ reaction.

## Mass photometry

The composition of the nanodisc-reconstituted protein preparation was assessed using mass photometry on a Refeyn OneMP instrument (Refeyn Ltd.), which was calibrated using an unstained native protein ladder (NativeMark Unstained Protein Standard A, Thermo Fisher Scientific Inc). Measurements were performed on the reconstituted complex I at a concentration of 0.015 mg ml$^{-1}$ using AcquireMP 2.2.0 software and were analyzed using the DiscoverMP 2.2.0 package.

## Preparation of cryo-EM samples

The cryo-EM samples were prepared using a CP3 cryoplunge (Gatan). Quantifoil R0.6/1 Cu300 holey carbon grids were cleaned with chloroform, acetone, and isopropanol as described by *Passmore and Russo, 2016*. The grids were glow discharged in the ELMO glow discharge system (Corduan Technologies) from both sides for 2 min at 11 mA and 0.28 mbar. For the nanodisc-

reconstituted sample, four microliters of the reconstituted protein solution at 0.15 mg ml$^{-1}$ concentration were applied on a grid and blotted from both sides for 2.2 s with Whatman No. 3 filter paper at 97% relative humidity. The LMNG-purified complex I was supplemented with 0.2% CHAPS (Anatrace) and applied at concentration 2–3 mg ml$^{-1}$. The grids were plunge-frozen in liquid ethane at −176°C and stored in liquid nitrogen.

## Cryo-EM data collection

Cryo-EM images were collected on a JEOL CryoARM 300 microscope equipped with an in-column Ω energy filter (*Fislage et al., 2020*) at 300 kV, automatically using SerialEM 3.0.8 (*Mastronarde, 2005*). The energy filter slit was set to 20 eV width. The nanodisc sample was collected at a nominal magnification of 60,000 and the corresponding calibrated pixel size of 0.771 Å. Five images per single stage position were collected using a cross pattern with three holes along each axis (*Efremov and Stroobants, 2021*). The 3 s exposures were dose-fractionated into 61 frames with an electron dose of 1.06 e- Å$^{-2}$ per frame. In total, 9122 zero-loss micrographs were recorded with the defocus varying between −0.9 and −2.2 μm (*Table 1*).

The LMNG-solubilized sample was collected at a 60,000 nominal magnification and the calibrated pixel size of 0.766 Å. Nine images were collected per stage position using a 3x3 hole pattern. The 3 s exposures were dose-fractionated into 60 frames with 1 e$^-$ A$^{-2}$ dose per frame. The defocus varied between −1.0 and −2.0 μm. During 36 hr of data collection, 13,084 zero-loss micrographs were recorded.

## EM image processing

For both datasets, the dose-fractionated movies were motion-corrected using MotionCor2 (*Zheng et al., 2017*) in the patch mode. The Contrast Transfer Function (CTF) parameters were estimated using CTFFIND-4.1 (*Rohou and Grigorieff, 2015*).

For processing of the nanodisc data, 40 micrographs of various defoci were selected, manually picked, and used to train the neural network of crYOLO 1.4 (*Wagner et al., 2019*). After training, 1,256,734 particles were picked automatically from the complete dataset, extracted in RELION 3.0 (*Zivanov et al., 2018*), and imported into cryoSPARC 2.11 (*Punjani et al., 2017*). Following 2D classification, six initial models were generated, among which one corresponded to the peripheral arm-only and another corresponded to the complete complex I. Using hetero-refinement, 441,265 and 525,680 particles were assigned to the peripheral arm and complete complex, respectively. Further processing was performed in RELION 3.1 (*Zivanov et al., 2020*). After per-particle CTF estimation and Bayesian polishing, 3D auto-refinement of the complete complex produced a map at an average resolution of 3.4 Å (*Figure 1—figure supplement 3*). However, the map was very heterogeneous with the peripheral arm resolved at 3.0–3.6 Å whereas the membrane arm was resolved at over 10 Å.

To address this heterogeneity, both arms were refined independently using multi-body refinement (*Nakane et al., 2018*; *Figure 1—figure supplement 3*) and the peripheral domain signal was subtracted. After two rounds of 3D classification applied to the membrane domain and nanodisc signal subtraction, a subset of 48,745 particles was 3D refined to an average resolution of 3.6 Å. However, the density map was anisotropic. To improve the reconstruction, the original stack of 525,680 particles was refined against the masked peripheral arm, followed by subtraction of the signal from the peripheral arm. Next, membrane arm map obtained above was filtered to 9 Å and used as an initial model for the 3D refinement of all resulting membrane arm particles. Next, to prevent model bias, the refined map was low-pass filtered to 20 Å and used in the subsequent 3D classification with 10 classes, τ of 12 and 24° local angular search range and 1.8° angular step. The best class (110,258 particles and 8 Å resolution) was auto-refined using the starting model low-pass filtered to 15 Å, which produced the reconstruction to a resolution of 4.4 Å. Next, the nanodisc density was subtracted, which further improved the resolution to 3.9 Å. Following 3D classification without alignment with τ of 40, eight classes, and resolution in the E-step limited to 4 Å, a subset of 37,441 particles was identified, which after auto refinement, produced a density map at an average resolution of 3.9 Å with better resolved peripheral regions. Finally, density modification with the resolve_cryo_em tool available in PHENIX 1.18.2 (*Terwilliger et al., 2020*) improved the resolution to 3.7 Å (*Figure 1—figure supplements 3* and *4*).

After multibody refinement of the arms described above, peripheral arm particles with the subtracted membrane arm were 3D classified into 12 classes without alignment using τ of 40 and resolution of the expectation step limited to 4 Å. The best class contained 134,976 particles and was further refined to 2.9 Å resolution.

A subset of 166,580 particles was selected after a similar 3D classification procedure that was applied to the 441,265 particles of dissociated peripheral arm particles. It was further cleaned from the remaining particles of the complete complex I by 2D classification, resulting in a subset of 151,357 particles that produced a density map to a resolution of 3.0 Å. As the reconstructions of the dissociated and membrane arm-subtracted peripheral arms were virtually identical, both stacks were combined. After two cycles of per-particle CTF refinement, aberration corrections, and Bayesian particle polishing in RELION 3.1, the resolution improved to 2.4 Å. Consecutive density modification in PHENIX further improved the resolution to 2.1 Å (*Figure 1—figure supplements 3* and *4*, *Table 1*).

To resolve the conformation of entire complex I, a stack of 525,680 particles was aligned to the peripheral arm using auto-refinement with a mask around the peripheral arm in RELION 3.1. Next, 3D classification without alignment into 30 classes with resolution of the expectation step limited to 20 Å and τ of 4 was performed, followed by auto-refinement of each resulting class, which produced maps to a resolution in the range of 9–20 Å (some of the classes are shown in *Figure 1—figure supplement 5*).

Three high-resolution conformations of complete complex I were obtained as follows. Conformation one was resolved by applying the 3D classification into 15 classes, τ of 6, a 24° local angular search range, and 1.8° sampling interval to the subset of 110,258 particles that produced the 3.9 Å reconstruction of the membrane arm (see above). The best class consisted of 23,445 particles that were refined to a resolution of 3.9 Å.

Conformations 2 and 3 were identified by applying 3D classification without image alignment into 12 classes with τ of 40 and resolution of the expectation step limited to 4 Å, to the stack of 525,680 intact complex I particles. Two of the best classes, consisting of 21,620 and 21,234 particles were refined to 4.6 Å and 4.5 Å, respectively. Following density-modification in PHENIX, the resolution of the maps was improved to 3.3, 3.8, and 3.7 Å, for conformations 1, 2, and 3, respectively (*Figure 1—figure supplement 4C*, *Table 1*).

From 13,084 motion-corrected micrographs containing LMNG-solubilized complex I, a subset of 9333 was selected. Next, 1,469,948 particles were picked with crYOLO 1.7.0. After 2D classification in cryoSPARC 3.2.0 792,120 particles were retained. The particles were subjected to 3D auto-refinement in RELION 3.1, followed by focused 3D auto-refinement with a mask around the peripheral arm. Next, focused 3D classification of the peripheral arm into 12 classes with τ = 6, 24° local angular search range and 1.8° angular step resulted in a homogeneous subset of 82,433 particles that after 3D refinement produced a 4.3 Å peripheral arm reconstruction. This stack was used for 3D classification without alignment into 15 classes with τ = 15 and without a mask. 3D auto-refinement of the resulting classes converged to resolution between 6.7–13 Å and revealed large relative movements of the arms (*Figure 6—figure supplement 3*).

## Model building

Peripheral arm subunits constituting NuoB, CD, E, F, G, and I were first homology modeled in the SWISS-MODEL server (*Waterhouse et al., 2018*) based on the structure of *T. thermophilus* (PDB ID:4HEA *Baradaran et al., 2013*) and were rigid-body fitted into the density map in UCSF Chimera 1.13.1 (*Pettersen et al., 2004*). Following manual rebuilding in Coot 0.9 (*Casañal et al., 2020*), the model was subjected to real-space refinement against the final 2.1 Å map of the peripheral arm in PHENIX 1.19.2 (*Liebschner et al., 2019*) using the default parameters. Secondary structure restrains were applied only to the interfacial region resolved at a lower resolution. The value of the nonbonded_weight parameter was optimized. Water molecules were added to the map and validated using the 'Check/delete waters' tool in Coot 0.9. Molecular dynamics-based model idealization was conducted in ISOLDE 1.0b5 (*Croll, 2018*), followed by several iterations of real-space refinement without atomic displacement parameter (ADP) restraints and manual rebuilding in Coot 0.9.

For the membrane domain, the previously obtained *E. coli* model (PDB ID: 3RKO) was real-space-refined in PHENIX. The missing NuoH subunit was homology-modeled using the *T. thermophilus* structure (PDB ID: 4HEA) in Coot 0.9. The final model was obtained after several rounds of manual rebuilding and real-space refinement using standard parameters with Ramachandran restrains,

secondary-structure restrains applied to the NuoL TMH9-13, without ADP restrains, and with the optimized nonbonded_weight parameter. To generate the model of the complete complex I, the separate peripheral and membrane arm structures were combined and the missing parts at the inter-face (*Table 2*) were built manually. As the density of NuoL, NuoM and NuoN was very poor in all the resolved full conformations, these subunits were refined as rigid-body in PHENIX, whereas the others were refined using real-space refinement with minimization_global, local_grid_search, morphing, and ADP refinement. Ramachandran, ADP, and secondary-structure restrains were used. The models were validated in MolProbity (*Williams et al., 2018*). Structural conservation was evaluated using the ConSurf server (*Ashkenazy et al., 2016*). The figures and videos were generated in UCSF ChimeraX version 1.1. (*Goddard et al., 2018*) and PyMOL (The PyMOL Molecular Graphics System, Version 2.4.1 Schrödinger, LLC).

## Acknowledgements

We are indebted to Henri De Greve for help with establishing CRISPR-Cas9 for *E. coli* complex I. We are thankful to Dr. Adam Schröfel and Dr. Marcus Fislage for providing support during cryo-EM data collection, to Annelore Stroobants for technical assistance and to Lukasz Milewski for assistance in data processing. We kindly thank VIB Tech Watch fund for facilitating access to the Refeyn instrument. We would like to acknowledge the funding provided by Vlaams Instituut voor Biotechnologie, Fonds Wetenschappelijk Onderzoek and by the European Research Council.

## Additional information

### Funding

| Funder | Grant reference number | Author |
|---|---|---|
| Fonds Wetenschappelijk On-derzoek | G0H5916N | Rouslan G Efremov |
| Fonds Wetenschappelijk On-derzoek | G.0266.15N | Rouslan G Efremov |
| H2020 European Research Council | 726436 | Rouslan G Efremov |

The funders had no role in study design, data collection and interpretation, or the decision to submit the work for publication.

### Author contributions

Piotr Kolata, Resources, Formal analysis, Validation, Investigation, Visualization, Methodology, Writing - review and editing; Rouslan G Efremov, Conceptualization, Resources, Formal analysis, Supervision, Funding acquisition, Validation, Writing - original draft, Project administration, Writing - review and editing

### Author ORCIDs

Piotr Kolata https://orcid.org/0000-0002-9484-5025
Rouslan G Efremov https://orcid.org/0000-0001-7516-8658

### Decision letter and Author response

Decision letter https://doi.org/10.7554/eLife.68710.sa1
Author response https://doi.org/10.7554/eLife.68710.sa2

## Additional files

### Supplementary files

• Transparent reporting form

## Data availability

Cryo-EM density maps and atomic models are deposited into the PDB and EMDB databases with the following accession codes: cytoplasmic domain (PDB ID: 7NZ1, EMD-12661), membrane domain (PDB ID: 7NYH, EMD-12652), entire complex conformation 1 (PDB ID: 7NYR, EMD-12653), conformation 2 (PDB ID: 7NYU,EMD-12654), conformation 3 (PDB ID: 7NYV, EMD-12655), reconstruction of entire complex I solubilised in LMNG (EMD-13291).

The following datasets were generated:

| Author(s) | Year | Dataset title | Dataset URL | Database and Identifier |
|---|---|---|---|---|
| Kolata P, Efremov RG | 2021 | Respiratory Complex I from Escherichia coli solubilised in LMNG | https://www.ebi.ac.uk/pdbe/entry/emdb/EMD-13291 | Electron Microscopy Data Bank, EMD-13291 |
| Kolata P, Efremov RG | 2021 | Respiratory complex I from *Escherichia coli* - focused refinement of cytoplasmic arm | https://www.ebi.ac.uk/pdbe/entry/emdb/EMD-12661 | Electron Microscopy Data Bank, EMD-12661 |
| Kolata P, Efremov RG | 2021 | Respiratory complex I from *Escherichia coli* - focused refinement of membrane arm | https://www.ebi.ac.uk/pdbe/entry/emdb/EMD-12652 | Electron Microscopy Data Bank, EMD-12652 |
| Kolata P, Efremov RG | 2021 | Respiratory complex I from *Escherichia coli* - conformation 1 | https://www.ebi.ac.uk/pdbe/entry/emdb/EMD-12653 | Electron Microscopy Data Bank, EMD-12653 |
| Kolata P, Efremov RG | 2021 | Respiratory complex I from *Escherichia coli* - conformation 2 | https://www.ebi.ac.uk/pdbe/entry/emdb/EMD-12654 | Electron Microscopy Data Bank, EMD-12654 |
| Kolata P, Efremov RG | 2021 | Respiratory complex I from *Escherichia coli* - conformation 3 | https://www.ebi.ac.uk/pdbe/entry/emdb/EMD-12655 | Electron Microscopy Data Bank, EMD-12655 |
| Kolata P, Efremov RG | 2021 | Respiratory complex I from *Escherichia coli* - focused refinement of cytoplasmic arm | https://www.rcsb.org/structure/7NZ1 | RCSB Protein Data Bank, 7NZ1 |
| Kolata P, Efremov RG | 2021 | Respiratory complex I from *Escherichia coli* - focused refinement of membrane arm | https://www.rcsb.org/structure/7NYH | RCSB Protein Data Bank, 7NYH |
| Kolata P, Efremov RG | 2021 | Respiratory complex I from *Escherichia coli* - conformation 1 | https://www.rcsb.org/structure/7NYR | RCSB Protein Data Bank, 7NYR |
| Kolata P, Efremov RG | 2021 | Respiratory complex I from *Escherichia coli* - conformation 2 | https://www.rcsb.org/structure/7NYU | RCSB Protein Data Bank, 7NYU |
| Kolata P, Efremov RG | 2021 | Respiratory complex I from *Escherichia coli* - conformation 3 | https://www.rcsb.org/structure/7NYV | RCSB Protein Data Bank, 7NYV |

The following previously published datasets were used:

| Author(s) | Year | Dataset title | Dataset URL | Database and Identifier |
|---|---|---|---|---|
| Baradaran R, Berrisford JM, Minhas GS, Sazanov LA | 2013 | Crystal structure of the entire respiratory complex I from Thermus thermophilus | https://www.rcsb.org/structure/4HEA | RCSB Protein Data Bank, 4HEA |
| Efremov RG, Sazanov LA | 2011 | Crystal structure of the membrane domain of respiratory complex I from E. coli at 3.0 angstrom resolution | https://www.rcsb.org/structure/3RKO | RCSB Protein Data Bank, 3RKO |

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

# Appendix 1

## Hypothetical mechanism of complex I and evolutionary-related complexes

The proposed mechanism is based on two simple principles: (1) the cavity for ubiquinone, or more generally proton acceptor, is sealed for external protons meaning that electron acceptors/donors can enter the cavity only in neutral form; and (2) the protons are delivered to/out of the cavity from extracellular space through subunit NuoH as shown in *Figure 7*.

In all solved structures of complex I and evolutionary-related hydrogenases the arms are bound tightly through a conserved interface. This is particularly pronounced in the structures of membrane bound hydrogenases where the cavity formed between peripheral and membrane domains is completely sealed (*Yu et al., 2020*; *Yu et al., 2018*). The necessity of having a tightly coupled cavity explains the high conservation of the subunit interface. Notably, the ubiquinone entrance is situated in the hydrophobic region of the bilayer, suggesting that charged molecules need to go through hydrophobic environment to reach the cavity, which is associated with very high energetic costs, hence even in the presence of such an opening the Q-cavity remains inaccessible to the charges.

Once bound to the cavity, ubiquinone can exchange electrons with cluster N2. It is well documented that the potential of benzoquinone-hydroquinone couple depends on the pH (*Chambers, 1988*), like that of any redox reaction involving protons, and the potential was shown experimentally to decrease by over 400 mV to below $-300$ mV upon pH change from 2 to 13 (*Gordillo and Schiffrin, 1994*; *Lemmer et al., 2011*).

Back of the envelope calculations show that the addition or extraction of a single proton from a cavity with the characteristic dimensions of the Q-cavity alters the activity of protons within the cavity by an equivalent of hundreds of millivolts. Thus, the redox potential of ubiquinone bound within the cavity enclosed from the environment is strongly modulated by the extraction/addition of single protons from/to the cavity. *Vice versa*, reduction or oxidation of ubiquinone/ubiquinol is equivalent to adding/removing proton binding groups to/from the Q-cavity. In this way, ubiquinone serves as a transformer that converts the energy of electrons into the chemical potential of protons in a fully reversible manner. During the forward cycle, ubiquinone reduction decreases proton activity in the cavity, which is rectified by protons entering the cavity and performing work. The question is where do the protons come from and how do they perform the work?

Multiple proton pathways have been suggested in the past (*Baradaran et al., 2013*; *Efremov and Sazanov, 2012*; *Kampjut and Sazanov, 2020*; *Verkhovskaya and Bloch, 2013*; *Yu et al., 2020*; *Yu et al., 2018*). However, they all end up on the intracellular/matrix side of the membrane, which makes it difficult to explain the energy conversion. Instead, we propose that the protons re-protonate ubiquinone through NuoH and/or adjacent trans-membrane subunits from the periplasm as shown in *Figure 7*. The fold of subunit NuoH contains a set of 5 TMH structurally similar to the symmetric module of antiporter-like subunits (*Baradaran et al., 2013*) such that invariant $^H$Glu157 superposes with $^M$Glu144.

We suggest that proton transport through NuoH is coupled to the transport of three protons by the three antiporter-like subunits in the opposite direction such that the entire membrane arm functions as a proton antiporter with a stoichiometry of $1H^+_{in}/3H^+_{out}$. The coupling likely proceeds through a classical alternating access mechanism (*Jardetzky, 1966*) that involves both the interaction of ionizable residues in the middle of the membrane (*Baradaran et al., 2013*; *Efremov and Sazanov, 2011*) and conformational changes. Four protons are translocated outside in two pumping cycles per one reduced ubiquinone molecule (*Figure 7*).

In the proposed model only the equilibrium potentials of NADH, ubiquinone, and electrochemical membrane potential are important for the directionality of the reaction and energy balance as expected for a molecular machine (*Astumian et al., 2016*). Using the proposed model, the activity of protons in the Q-cavity under equilibrium condition can be easily calculated using Nernst equation with two assumptions: (1) protons reach cavity only from outer space and (2) the electric component of the electrochemical potential on the membrane is the same throughout the membrane including subunit NuoH. Then net reaction facilitated by a generalized complex functioning by the proposed mechanism and translocating n protons through antiporter-like subunits (n=3 for complex I) can be written as follows:

$$NADH + UQ + 2H^+_{cav} \leftrightarrow NAD^+ + H^+_{in} + UQH_2 \qquad (1)$$

$$2nH^+_{in} + 2H^+_{out} \leftrightarrow 2nH^+_{out} + 2H^+_{cav} \qquad (2)$$

The *Equation 1* describes redox reaction and the *Equation 2* vectorial proton transport. Here $H^+_{cav}$ refers to the protons in the Q-cavity.

The Gibbs free energy of the net reaction is given by the expression:

$$\Delta G = -2F(E_{0NADH} + E_{0UQ}) + RTln\frac{[NAD^+][UQH_2][H^+_{out}]^{2(n-1)}}{[NADH][Q][H^+_{in}]^{2n-1}} - 2(n-1)F\Delta\Psi \qquad (3)$$

, where $\Delta\Psi$ is an electrical component of the membrane potential, $E_{0\ NADH}$ and $E_{0\ UQ}$ are standard potentials for NADH and UQ, R is gas constant and F is Faraday constant.

Using Nernst equation, the apparent equilibrium pH in the Q-cavity can be calculated as:

$$pH_{cav} = pH_{out} + n(pH_{in} - pH_{out}) - \frac{(n-1)F\Delta\Psi}{2.3RT} \qquad (4)$$

The values of $pH_{cav}$ calculated for different n are summarized in *Appendix 1—table 1*. They show that under equilibrium conditions in complex I (n=3) $pH_{cav}$ is very basic suggesting that a significant fraction of dianion $UQ^{2-}$ will be present under equilibrium conditions. Under these conditions, redox potential of UQ is low and comparable to that of NADH (−320 mV) suggesting that electrons do not 'lose' energy upon transfer from NADH to ubiquinone. The mechanism is consistent with the appearance of trans-membrane potential-dependent semiquinone species observed by EPR spectroscopy in tightly coupled submitochondrial particles (*Yano et al., 2005*).

**Appendix 1—table 1.** Equilibrium $pH_{cav}$ and proton translocation stoichiometry for various values of n as defined in the Appendix and calculated for $\Delta\Psi$ −120 mV, ΔpH 0.9 and matrix $pH_{out}$ 7.4 characteristic of mitochondria (*Mollica et al., 1998*).

| N | pH$_{cav}$ | Stoichiometry xH$^+$/2e$^-$ |
|---|---|---|
| 0 | 5.4 | -2 |
| 1 | 8.3 | 0 |
| 2 | 11.2 | 2 |
| 3 | 14.1 | 4 |

The proposed mechanism is applicable to the hydrogenases and oxidoreductases evolutionary related to complex I (*Efremov and Sazanov, 2012*; *Yu et al., 2020*; *Yu et al., 2018*). In all of them, the cavity formed between the peripheral arm and NuoH homologs is sealed. According to our model, membrane-bound sulfane sulfur reductase (MBS) corresponds to n=2 and may function as a proton pump (assuming it does not translocate other ions) with a stoichiometry of 2H$^+$/2e$^-$ (*Appendix 1—table 1*). More interesting are the cases of n=1 and 0. The case of n=1 corresponds to membrane-bound hydrogenase (MBH). If the antiporter-like subunit of MBH translocates a single proton, then the complex is not electrogenic. However, since both MBH and MBS are different from complex I in the orientation of their antiporter-containing module (*Yu et al., 2020*; *Yu et al., 2018*) and are closer in structure to Mrp antiporter (*Steiner and Sazanov, 2020*), it is possible that they translocate monovalent metal ions. In this case, MBH can still couple redox reaction to the generation of electrochemical potential by coupling proton transport inside to Na$^+$ translocation outside the cell. The case of n=0 corresponds to a complex of the peripheral arm with NuoH. Such a sub-complex is undoubtedly one of the stand-alone evolutionary modules. This suggestion is supported by the differences in its position between complex I and MBH/MBS (*Yu et al., 2020*; *Yu et al., 2018*) and its susceptibility to dissociation from the membrane arm in *E. coli* (*Baranova et al., 2007*; *Efremov and Sazanov, 2011*) as is expected for a late evolutionary addition (*Levy et al., 2008*). Such a complex dissipates proton potential or generates it if the redox reaction is reversed. We can speculate that the initial association of the hydrogen-evolving module with an antiporter may have

had an evolutionary advantage with the proton-translocating module serving as a source of protons (*Yu et al., 2018*) biasing $H_2$ evolution towards the reaction product (*Boyd et al., 2014*), as follows from *Equation 3* and *Appendix 1—table 1*.

Experiments with engineering *E. coli* complex I lacking subunit NuoL and *Y. lipolytica* complex I lacking homologs of subunits NuoM and NuoL (*Dröse et al., 2011*; *Steimle et al., 2011*) (correspond to n=2 and 1, respectively) both suggested that the engineered complexes were active and for both constructs stoichiometry was estimated as $2H^+/2e^-$. While NuoL deletion experiments support our model, the NuoL/M deletion clearly contradicts it. Both experiments should be interpreted cautiously, however. Results of NuoL deletion for *E. coli* complex I were not reproducible (*Verkhovskaya and Bloch, 2013*). In the case of *Y. lipolytica*, the homologs of NuoL/M dissociated from the complex along with another 11 subunits upon deletion of supernumerary subunit NB8M located at the tip of NuoL (*Zickermann et al., 2015*). Since the proton-translocating modules were not deleted per se, the presence of contaminating amounts of assembled complex I in the preparations that generated observed proton pumping cannot be completely excluded. It is important to note that mutation of the conserved ionizable residues on the interface between NuoN and NuoM, i.e. $^M$E144 (*Torres-Bacete et al., 2007*) or its counter ion $^N$K395 (*Amarneh and Vik, 2003*), result in a completely inactive complex I suggesting that dissociation of subunits NuoL/M also should render complex I inactive (*Verkhovskaya and Bloch, 2013*).

From our model it follows that complex uncoupling is achieved by opening the Q-cavity to the solvent. It is consistent with an elegant experiments by *Cabrera-Orefice et al., 2018* in which locking the conserved $^A$TMH1-2 plug with the cysteine bridge reversibly uncoupled the enzyme. Close examination of the crosslinked structure indicates that crosslinking fixes the plug in a conformation rendering the Q-cavity accessible to the solvent.

The exact proton translocation mechanism within the antiporter-like module is unknown and requires further experimental and computational modeling. Here, we can only speculate that given the high conservation of $^H$Glu157, it plays an important role in the coupling and may change its protonation state during the pumping cycle. Thus, it can influence the $pK_a$ of neighboring ionizable residues. In MBH, an equivalent $^M$Glu141 is separated from the closest ionizable $^H$Lys409 by distance of 20 Å, which in a hydrophobic environment with a dielectric constant of 10, allow them to mutually modulate the pKa of each other by approximately 1 pH unit, similar to the free energy conserved upon ferredoxin oxidation by the protein complex. This distance is reduced to around 13 Å in MBS and to around 6 Å in complex I, consistent with the proportionally higher free energy of catalyzed reactions. While many questions on the coupling mechanism of complex I remain open, further significant advances might be expected once structures of different conformation associated with proton translocation by membrane arm will be resolved.

