## [Decision Letter]

**Acceptance summary:**

The manuscript reports the cryoEM structure of a functional *E. coli* respiratory complex I (proton-pumping NADH-ubiquinone oxidoreductase) reconstituted in lipid nano-discs. The reconstructions and models presented by the authors indicate interesting *E. coli* specific features of the complex. Overall this can be a major advance for the structure of this important respiratory complex from a key model organism.

**Decision letter after peer review:**

Thank you for submitting your article "Structure of *Escherichia coli* respiratory complex I reconstituted into lipid nanodiscs reveals an uncoupled conformation" for consideration by *eLife*. Your article has been reviewed by 2 peer reviewers, and the evaluation has been overseen by a Reviewing Editor and Kenton Swartz as the Senior Editor. The following individuals involved in review of your submission have agreed to reveal their identity: Vinoth Kumar (Reviewer #1); James A Letts (Reviewer #2).

Essential revisions:

1) Address the reviewers comments concerning the quality of the maps and models. You will find recommendations below, but the key is that the submitted pdb needs to reflect the density you obtain and the quality of the density needs to be clearly stated in the manuscript.

2) The model that is put forward is interesting but currently not well supported by the data in the paper. The authors need to either move the model discussion to supplementary or include data to support it.

*Reviewer #1 (Recommendations for the authors):*

The manuscript and some figures could be improved for better understanding and several points are given below.

1) Line 36/37 – sentence 'Mitochondrial complex I has molecular weight 1 MDa.…

'has a molecular weight of 1 MDa.., whereas bacterial analogues are much smaller with molecular weight of ~500 kDa'.

2) Line 55-57 – Two term is mentioned here Q-cavity and Q-site. I would assume that they are the same i.e., ubiquonine is bound in the cavity and it interacts with N2 cluster. If they are different then it needs to be clarified.

3) Line 88 – monodispersed to monodisperse.

4) The activity assay described (lines 90-95) is only for the electron transfer or reduction of ubiquinone and not the proton translocation. This should be made clear. The sentence 'Furthermore, NADH:Q1.…' says that complex I was intact and catalytically active in a detergent-free environment'. Not sure, detergent-free environment is needed. One would like to think that without detergent the activity should be better or equally good. Please rephrase.

5) Supplementary figure 2. MSP and ND are used (gel is labelled as MSP and legend has ND). Better to use one of them. Readers not familiar with the terminology might get confused.

6) Supplementary figure 4 – please add a scale bar for micrograph in panel A.

7) Figure 1 – legend, please add (transparent gray) after 'the nanodisc density'.

8) Line 116 – The RMSD atom counts are confusing. the comparison with Thermus has RMSD 6.3 Å over 2593, 4 Å for membrane arm over 1604 atoms and peripheral arm 855 but the number don't add (1604+855 = 2459). Similarly, for comparison with mammalian enzyme. Please check and also rephrase the sentence.

9) Figure 2 – legend has panel E but the figure has only four. The text has it right. Please correct this.

10) Line 127 – please also include Agip et al. 2018.

11) In the text, the description of the soluble domain has been called both peripheral and cytoplasmic arms/domain. It may be useful to keep them uniform.

12) The paragraph (line 212-217) describing the Q-site stands alone here. Initially, I thought it can be moved elsewhere but it will affect the flow. Instead, it will be useful to add a figure showing this site as a panel in figure 2. (Although, panel 5D shows the Q-cavity).

13) Line 243 – the reference to Figure 3A can be included here.

14) Line 281-282 – 'Simultaneously there is sufficient room to accommodate the TM helix within the nanodisc'. The sentence is not clear. Is this to reflect that the TMH1 is disordered due to no space in nanodisc or thinning of membrane. Along these lines, if the TMH1 is mobile, would signal subtraction and focused classification around this region help to understand the dynamics of this helix.

15) Lines 303-307 – Please rephrase the sentences, they are complex and confusing.

16) Figure 4C is referred in few different places but there is no panel in the figure. I guess it is 4B. Please check and correct. Please improve the panel B in figure 4. One suggestion – to show the *E. coli* map + model in one panel and the overlay of *E. coli* and Thermus in another.

17) For reconstitution of the enzyme into lipid nanodiscs, was extra lipids added or the lipid that was used during purification (0.003%) is the sole source of lipid. The purification of MSP2N2 by Girnkova et al. 2010 is mentioned but not sure they reconstituted complex I. Please add this clarification.

18) Supplementary figure 2 – the peak (green rectangle) was used for Mass photometry as well as cryoEM. In the mass photometry (panel C), there is still empty ND. Why this should be?

19) Phenix density modification for cryoEM is used in all the maps and there is an increase in resolution (along expected lines). Is this just on number or was there a real improvement in the quality of the maps. Will it be possible to show some examples of densities as a supplementary figure (as panels C and D in supp figure 6. The current map shown must be the den mod map?) of the final post-processed and den mod maps, in particular the distal end of membrane arm will be interesting (as the local resolution is poor around that region).

*Reviewer #2 (Recommendations for the authors):*

My major concern with the manuscript is the map model correlations. The authors need to improve the models to be more consistent with the data or more clearly justify why this is not possible and clearly indicate to non-experts in structural biology where the low confidence regions are located in the models. The best way to do this would be to make regions in the model in which the density is poor for side chains poly-alanine and delete regions in which no clear main chain density is visible. This is an important structure and will be used to guide future experiments, therefore it is important to ensure that the limitations of the structure are made clear.

---

## [Author Response]

Reviewer #1 (Recommendations for the authors):The manuscript and some figures could be improved for better understanding and several points are given below.1) Line 36/37 – sentence 'Mitochondrial complex I has molecular weight 1 MDa.…'has a molecular weight of 1 MDa.., whereas bacterial analogues are much smaller with molecular weight of ~500 kDa'

The language has been corrected.

2) Line 55-57 – Two term is mentioned here Q-cavity and Q-site. I would assume that they are the same i.e., ubiquonine is bound in the cavity and it interacts with N2 cluster. If they are different then it needs to be clarified.

We do make a distinction between Q-cavity, which refers to complete cavity formed between subunits NuoB/NuoD/NuoH, and Q-site which refers to the immediate environment of the ubiquinone head group, the region where electron transfer and protonation of ubiquinone takes place. In the revised manuscript we have improved the definitions on lines 55-58. Now, the text reads as follows:

“Together with subunits NuoB and NuoD it forms an extended ubiquinone-binding cavity (Q-cavity) that stretches from the hydrophobic region of the membrane bilayer to the binding site of the ubiquinone head group (Q-site) found in the proximity of the terminal iron-sulfur cluster N2 (Baradaran et al., 2013).”

3) Line 88 – monodispersed to monodisperse.

The language has been corrected.

4) The activity assay described (lines 90-95) is only for the electron transfer or reduction of ubiquinone and not the proton translocation. This should be made clear. The sentence 'Furthermore, NADH:Q1.…' says that complex I was intact and catalytically active in a detergent-free environment'. Not sure, detergent-free environment is needed. One would like to think that without detergent the activity should be better or equally good. Please rephrase.

The redox activity of complex I was remeasured, because of a flaw identified in the previously reported measurements (qualitatively results remained unchanged) and supplemented with NADH:DQ activity measurements. This resulted in altered Figure 1—figure supplement 2. We have rewritten the paragraph on lines 90-98 that now incorporates suggestions of the reviewer:

“Reconstituted complex I was active in catalyzing NADH:ubiquinone-1 (Q1) and NADH:decylubiquinone (DQ) redox reactions (Figure 1—figure supplement 2D, E). […] While without compartmentalization the proton translocation activity cannot be assessed, the lipid environment provided by the nanodisc is expected to mimic closely lipid vesicles in which reconstituted purified *E. coli* complex I was shown to pump protons (Steimle et al., 2011).”

5) Supplementary figure 2. MSP and ND are used (gel is labelled as MSP and legend has ND). Better to use one of them. Readers not familiar with the terminology might get confused.

The label of SDS-PAGE lane was changed from “MSP” to “ND”.

6) Supplementary figure 4 – please add a scale bar for micrograph in panel A.

The scale bar has been added to the figure.

7) Figure 1 – legend, please add (transparent gray) after 'the nanodisc density'.

The figure legend has been corrected as requested.

8) Line 116 – The RMSD atom counts are confusing. the comparison with Thermus has RMSD 6.3 Å over 2593, 4 Å for membrane arm over 1604 atoms and peripheral arm 855 but the number don't add (1604+855 = 2459). Similarly, for comparison with mammalian enzyme. Please check and also rephrase the sentence.

We have revised this paragraph and recalculated RMSD values without applying cutoff during alignment to obtain unbiased comparison. While RMSD values have increased, they include 85 to 90% of all Ca atoms of the corresponding structures and present a more objective comparison that covers nearly entire structure. The numbers of Ca atoms for cytoplasmic, membrane arms and entire complex now add up.

The text has been amended as follows (from line 122):

“The fold and arrangement of the *E. coli* complex I subunits is mainly similar to the structures of other complex I homologs Somewhat high values of RMSD obtained for structural alignments of the entire complexes or its individual arms to *Thermus thermophilus* [RMSD 7.3 Å (4199 Cα) for entire complex, 4.3 Å (2070 Cα) for membrane arm, and 8.1 Å (2129 Cα) for peripheral arm] and ovine enzyme [8.5 Å (4007 Cα) for entire complex, 5.0 Å (1969 Cα) for membrane arm, and 8.4 Å (2038 Cα) for the peripheral arm] (Figure 1B) reflect long-range twisting and bending of arms observed between complex I from different species (Agip et al., 2018; Baradaran et al., 2013; Vinothkumar et al., 2014).”

9) Figure 2 – legend has panel E but the figure has only four. The text has it right. Please correct this.

The figure legend has been corrected accordingly.

10) Line 127 – please also include Agip et al. 2018

The citation has been added on line 129.

11) In the text, the description of the soluble domain has been called both peripheral and cytoplasmic arms/domain. It may be useful to keep them uniform.

We replaced “cytoplasmic arm” by “peripheral arm” throughout the manuscript. While cytoplasmic arm is more accurate term for bacterial complex, peripheral arm is also applicable to the mitochondrial complex I in which the arm points into the mitochondrial matrix.

12) The paragraph (line 212-217) describing the Q-site stands alone here. Initially, I thought it can be moved elsewhere but it will affect the flow. Instead, it will be useful to add a figure showing this site as a panel in figure 2. (Although, panel 5D shows the Q-cavity).

A panel D showing Q-site has been added to the Figure 5.

13) Line 243 – the reference to Figure 3A can be included here.

The reference has been included (line 249) and the sentence has been rephrased:

“Cluster N3 interacts with *E. coli*-specific ^F^His400 (Figure 3A); however, the potential of N3 is very similar between the species …”

14) Line 281-282 – 'Simultaneously there is sufficient room to accommodate the TM helix within the nanodisc'. The sentence is not clear. Is this to reflect that the TMH1 is disordered due to no space in nanodisc or thinning of membrane. Along these lines, if the TMH1 is mobile, would signal subtraction and focused classification around this region help to understand the dynamics of this helix.

The confusion likely stems from the fact that by thickness we meant 2 different dimensions: 1) thickness in the direction perpendicular to the bilayer and 2) depth of the nanodisc density in the membrane plane. The depth of nanodisc is the same next to ^H^TMH1 as in the other regions of the membrane arm. Based on additional reconstructions obtained in detergent and taking into consideration the comments of referee #2, we revised this paragraph and moved it to the end of discussion and now on lines 427-433 it reads as follows:

“^H^TMH1 is exposed to the lipid environment and the width of the nanodisc next to ^H^TMH1 is similar to other regions around the membrane arm (Video 1). […] By comparing the detergent-solubilized and reconstituted complexes we can conclude that position and dynamics of this helix is neither the cause of the uncoupled conformation nor of the high relative mobility of the arms.”

To improve clarity we also modified description of the nanodisc density on line 217:

“The belt locally bends next to the subunit NuoL at the region where it interacts with the long amphipathic helix and amphipathic helix connecting ^H^TMH1-TMH2 (^H^AH1) protrudes into the nanodisc (Video 1)”.

As suggested by the reviewer, we performed focused 3D classification on the sub volume containing the missing helix. It did reveal some additional density, but the results were not very conclusive.

15) Lines 303-307 – Please rephrase the sentences, they are complex and confusing.

We rephrased these sentences and now they read as follows (line 300):

“Analysis of cavities and potential hydration sites using DOWSER++ (Morozenko and Stuchebrukhov, 2016) shows that the residues ^H^Glu157 and ^A^Asp79 along with the carbonyl oxygen of ^J^Gly61 on the p-bulge of ^J^TM3 (Efremov and Sazanov, 2011) point into a hydrophilic cavity that can accommodate several water molecules enabling proton exchange between the ionizable residues. Similarly, cavities that can be hydrated link a chain of ionizable residues ^J^Glu55-^K^Glu36-^K^Glu72-^N^Glu133 potentially enabling proton exchange between its ends.”

16) Figure 4C is referred in few different places but there is no panel in the figure. I guess it is 4B. Please check and correct. Please improve the panel B in figure 4. One suggestion – to show the *E. coli* map + model in one panel and the overlay of *E. coli* and Thermus in another.

The figure has been corrected and panel B has been split into two images as suggested by the reviewer. References to Figure 4C were replaced with references to Figure 4B.

17) For reconstitution of the enzyme into lipid nanodiscs, was extra lipids added or the lipid that was used during purification (0.003%) is the sole source of lipid. The purification of MSP2N2 by Girnkova et al. 2010 is mentioned but not sure they reconstituted complex I. Please add this clarification.

Girnkova et al. 2010 describe production of MSP2N2 protein but not the reconstitution of complex I into lipid nanodiscs. To improve clarity, we removed the reference to Girnkova et al. 2010 from the Results section. Girnkova et al. 2010 is still referenced in Materials and methods where it is clearly refers to production of MSP2N2 (line 524):

“The membrane scaffold protein MSP2N2 was expressed and purified following a published protocol (Grinkova et al., 2010).”

During the reconstitution no addition lipids were added. This is now clarified as follows (line 525):

“Purified, lipid-containing complex I preparation at 520 nM concentration was mixed with 10.4 µM MSP2N2 (1:20 protein:MSP molar ratio, no additional lipids were added during reconstitution) and incubated for 1 hour at 4°C.”

The optimal amount of MSP2N2 was determine empirically because lipids concentrate together with detergent during protein concentration and their final concentration was not known. Therefore, the ratio is given as protein:MSP rather than lipid:MSP ratio. Assuming that most of lipids are retained by concentrator, the estimated lipid:MSP molar ratio is around 135:1.

18) Supplementary figure 2 – the peak (green rectangle) was used for Mass photometry as well as cryoEM. In the mass photometry (panel C), there is still empty ND. Why this should be?

Reconstitution of complex I into lipid nanodiscs resulted in around 10-fold molar excess of empty nanodiscs as compared to those containing complex I. Due to nanodiscs heterogeneity, subsequent separation by gel filtration had a limited efficiency and some larger, empty nanodiscs were still present in the final protein preparation. We now mention this in the legend of Figure 1—figure supplement 2:

“(C) Mass photometry of the reconstituted complex I pooled form the main gel filtration peak. Left: a representative mass histogram showing two main peaks: ‘Empty’ nanodiscs at 95 kDa that co-elutes with complex I during size-exclusion chromatography and the nanodisc-reconstituted complex I at 713 kDa.”

19) Phenix density modification for cryoEM is used in all the maps and there is an increase in resolution (along expected lines). Is this just on number or was there a real improvement in the quality of the maps. Will it be possible to show some examples of densities as a supplementary figure (as panels C and D in supp figure 6. The current map shown must be the den mod map?) of the final post-processed and den mod maps, in particular the distal end of membrane arm will be interesting (as the local resolution is poor around that region).

In our experience, success of phenix density modification is reconstruction-depended. For our complex I maps, nominal resolution has improved by a few tenth of Å after density modification (Figure 1—figure supplement 4). The density quality improved as well. It was particularly spectacular for high-resolution map of peripheral arm, but less efficient in the regions resolved at lower resolution. To illustrate it, we added additional panels to the Figure 1—figure supplement 6 that show maps before and after the modification.

To clarify that density modified maps were used, we mentioned it on the lines 109 and 115 of the Results section in the revised manuscript and in the legend of Figure 1—figure supplement 4.

The RELION non-modified half-maps and masks used for Post Processing in RELION were deposited to EMDB for each reconstruction along with the density-modified map. This will allow readers to do a thorough comparison of density modified map with simply sharpened maps.

The model-map Fourier shell correlation of the refined models is better for the density-modified maps (as shown in Author response image 1). Moreover, refinement of models against density modified maps can slightly improve the model correlation with non-density sharpened map.

**Author response image 1. sa2fig1:** Figure shows model-map FSC curves between the models refined against RELION-sharpened or density-modified maps (PDB: Relion or Phenix_resolve, respectively) and maps sharpened in RELION or density-modified (Map: Relion or Phenix_resolve, respectively). The FSC curves are shown for the membrane arm (A) and peripheral arm (B) models.

Reviewer #2 (Recommendations for the authors):My major concern with the manuscript is the map model correlations. The authors need to improve the models to be more consistent with the data or more clearly justify why this is not possible and clearly indicate to non-experts in structural biology where the low confidence regions are located in the models. The best way to do this would be to make regions in the model in which the density is poor for side chains poly-alanine and delete regions in which no clear main chain density is visible. This is an important structure and will be used to guide future experiments, therefore it is important to ensure that the limitations of the structure are made clear.

Real space correlation strongly depends on the way the density is filtered and is particularly affected by the single-to-noise ratio of the map coefficients. Due to the weaker maps and their heterogeneity, the cross-correlation coefficients are not very high, but they are nonetheless above the acceptable cutoff of 0.5.

As requested, the models have been modified and the side chains without density have been cut from the models of the membrane and peripheral arms, as well as from the full conformations.